# Ketogenic diet restrains aging-induced exacerbation of coronavirus infection in mice

Seungjin Ryu[1,2], Irina Shchukina[3], Yun-Hee Youm[1,2], Hua Qing[4], Brandon Hilliard[4], Tamara Dlugos[1,2], Xinbo Zhang[1], Yuki Yasumoto[1], Carmen J Booth[1], Carlos Fernández-Hernando[1,5], Yajaira Suárez[1,5], Kamal Khanna[6], Tamas L Horvath[1,5,7], Marcelo O Dietrich[1,5], Maxim Artyomov[3], Andrew Wang[2,4]*, Vishwa Deep Dixit[1,2,5,7]*

[1]Department of Comparative Medicine, Yale School of Medicine, New Haven, United States; [2]Department of Immunobiology, Yale School of Medicine, New Haven, United States; [3]Department of Pathology and Immunology, Washington University School of Medicine, St. Louis, United States; [4]Department of Internal Medicine, Yale School of Medicine, New Haven, United States; [5]Program in Integrative Cell Signaling and Neurobiology of Metabolism, Yale School of Medicine, New Haven, United States; [6]Department of Microbiology, New York University Langone Health, New York, United States; [7]Yale Center for Research on Aging, New Haven, United States

*For correspondence:
andrew.wang@yale.edu (AW);
vishwa.dixit@yale.edu (VDD)

Competing interests: The authors declare that no competing interests exist.

**Abstract** Increasing age is the strongest predictor of risk of COVID-19 severity and mortality. Immunometabolic switch from glycolysis to ketolysis protects against inflammatory damage and influenza infection in adults. To investigate how age compromises defense against coronavirus infection, and whether a pro-longevity ketogenic diet (KD) impacts immune surveillance, we developed an aging model of natural murine beta coronavirus (mCoV) infection with mouse hepatitis virus strain-A59 (MHV-A59). When inoculated intranasally, mCoV is pneumotropic and recapitulates several clinical hallmarks of COVID-19 infection. Aged mCoV-A59-infected mice have increased mortality and higher systemic inflammation in the heart, adipose tissue, and hypothalamus, including neutrophilia and loss of γδ T cells in lungs. Activation of ketogenesis in aged mice expands tissue protective γδ T cells, deactivates the NLRP3 inflammasome, and decreases pathogenic monocytes in lungs of infected aged mice. These data establish harnessing of the ketogenic immunometabolic checkpoint as a potential treatment against coronavirus infection in the aged.

## Introduction

Aging-driven reduced resilience to infections is dependent in part on the restricted T cell repertoire diversity together with impaired T and B cell activation as well as inflammasome-driven low-grade systemic inflammation that compromises innate immune function (*Akbar and Gilroy, 2020*; *Camell et al., 2017*; *Youm et al., 2013*). Consequently, 80% of deaths due to COVID-19 in USA are in adults >65 years old (https://www.cdc.gov/), and aging is the strongest factor to increase infection fatality (*Pastor-Barriuso et al., 2020*; *Perez-Saez et al., 2021*; *Ward et al., 2020*). Lack of an aging animal model that mimics SARS-CoV-2 immunopathology has been a major limitation in the effort to determine the mechanism of disease and to develop effective therapeutics for the elderly. Inability of mouse ACE2 to bind SARS-CoV-2 is a significant hurdle in understanding the basic mechanism of

COVID-19. Accordingly, several approaches have been employed to develop models including introduction of human-ACE2 in mice and transient induction of hACE2 through adenoviral-associated vectors. These models have begun to yield important information on the mechanism of disease development. For example, epithelial cell-specific induction of hACE2 (K18-hACE2) as a model of SARS-CoV-2 infection demonstrated that post-intranasal inoculation, animals develop lung inflammation and pneumonia driven by infiltration of monocytes, neutrophils, and T cells (*Winkler et al., 2020*). Also, initial studies that employ lung-ciliated epithelial cell-specific HFH4/FOXJ1 promoter-driven hACE2 transgenic mice show that SARS-CoV-2 infection induces weight loss, lung inflammation, and approximately 50% mortality rate, suggesting the usefulness of this model to understand the mechanism of immune dysregulation (*Jiang et al., 2020*). However, significant hurdles remain to understand the mechanism and test therapeutic interventions that are relevant to disease severity in elderly, as complicated breeding and specific mutations need to be introduced in hACE2 transgenic strains in addition to the time required to age these models. The mouse model of SARS-CoV-2 based on adeno-associated virus (AAV)–mediated expression of hACE2 may allow circumvention of the above constrains. The delivery of hACE2 into the respiratory tract of C57BL/6 mice with AAV9 causes a productive infection as revealed by >200-fold increase in SARS-CoV-2 RNA and shows similar interferon gene expression signatures as COVID-19 patients (*Israelow et al., 2020*). However, in young wild-type mice, this model induces mild acute respiratory distress syndrome (ARDS) and does not cause neutrophilia, weight loss, or lethality (*Israelow et al., 2020*). Other studies using replication-deficient adenovirus-mediated transduction of hACE in mice and infection with SARS-CoV-2 produced 20% weight loss including lung inflammation (*Hassan et al., 2020*; *Sun et al., 2020*). Furthermore, genetic remodeling of the SARS-CoV-2 spike receptor binding domain that allow interaction with mACE demonstrated peribronchiolar lymphocytic inflammatory infiltrates and epithelial damage but no weight loss in infected mice (*Dinnon et al., 2020*). Moreover, middle-aged female mice that are 1 year old, (analogous to approximately 43 year old human), display greater lung pathology and loss of function post-infection with 10% weight loss followed by spontaneous recovery 7 days post-infection (*Dinnon et al., 2020*).

The mouse hepatitis virus (MHV) and SARS-CoV-2 are both ARDS-related beta coronaviruses with a high degree of homology (*Gorbalenya et al., 2020*). Importantly, mCoV-A59 utilizes the entry receptor CEACAM1, which is expressed not only on respiratory epithelium, but also on enterocytes, endothelial cells, and neurons, much like ACE2 (*Godfraind et al., 1995*), thus allowing the study of wide-ranging systemic impacts of infection. The MHV infection is known to cause hepatitis and encephalomyelitis (*Lavi et al., 1984*). Importantly, however, the intranasal infection with mCoV-A59 is pneumotropic and causes ARDS in C57BL/6J animals, while all other MHV strains require the A/J or type-I interferon-deficient background, for the development of severe disease (*De Albuquerque et al., 2006*; *Khanolkar et al., 2009*; *Yang et al., 2014*) limiting their use.

Aging-induced chronic inflammation in the absence of overt infections is predominantly driven by the NLRP3 inflammasome (*Bauernfeind et al., 2016*; *Camell et al., 2017*; *Youm et al., 2013*), a myeloid cell-expressed multiprotein complex that senses pathogen-associated molecular patterns (PAMPs) and danger-associated molecular patterns (DAMPs) to cause the processing and secretion of IL-1β and IL-18. There is increasing evidence that SARS-CoV-2 infection activates the NLRP3 inflammasome with increased levels of IL-18 and lactate dehydrogenase (LDH) levels due to inflammasome-mediated pyroptotic cell death (*Lucas et al., 2020*; *Zhou et al., 2020*). It is now known that increased glycolysis, which activates inflammasome, is associated with worsened COVID-19 outcome (*Codo et al., 2020*). This raises the question whether the substrate switch from glycolysis-to-ketogenesis can be employed to stave off COVID-19 in high-risk elderly population. Here, we establish that intranasal infection with mCoV-A59 recapitulates clinical features of COVID-19 seen in elderly and demonstrate that ketone metabolites protect against disease through inhibition of NLRP3 inflammasome and expansion of protective γδ T cells in lungs.

## Results

### mCoV-A59 infection in aged mice mimics COVID-19 severity

To determine the underlying deficits in immune and inflammatory response in aging, we investigated the impact of mCoV-A59 intranasal inoculation on adult (2–6 months) and old male mice (20–24

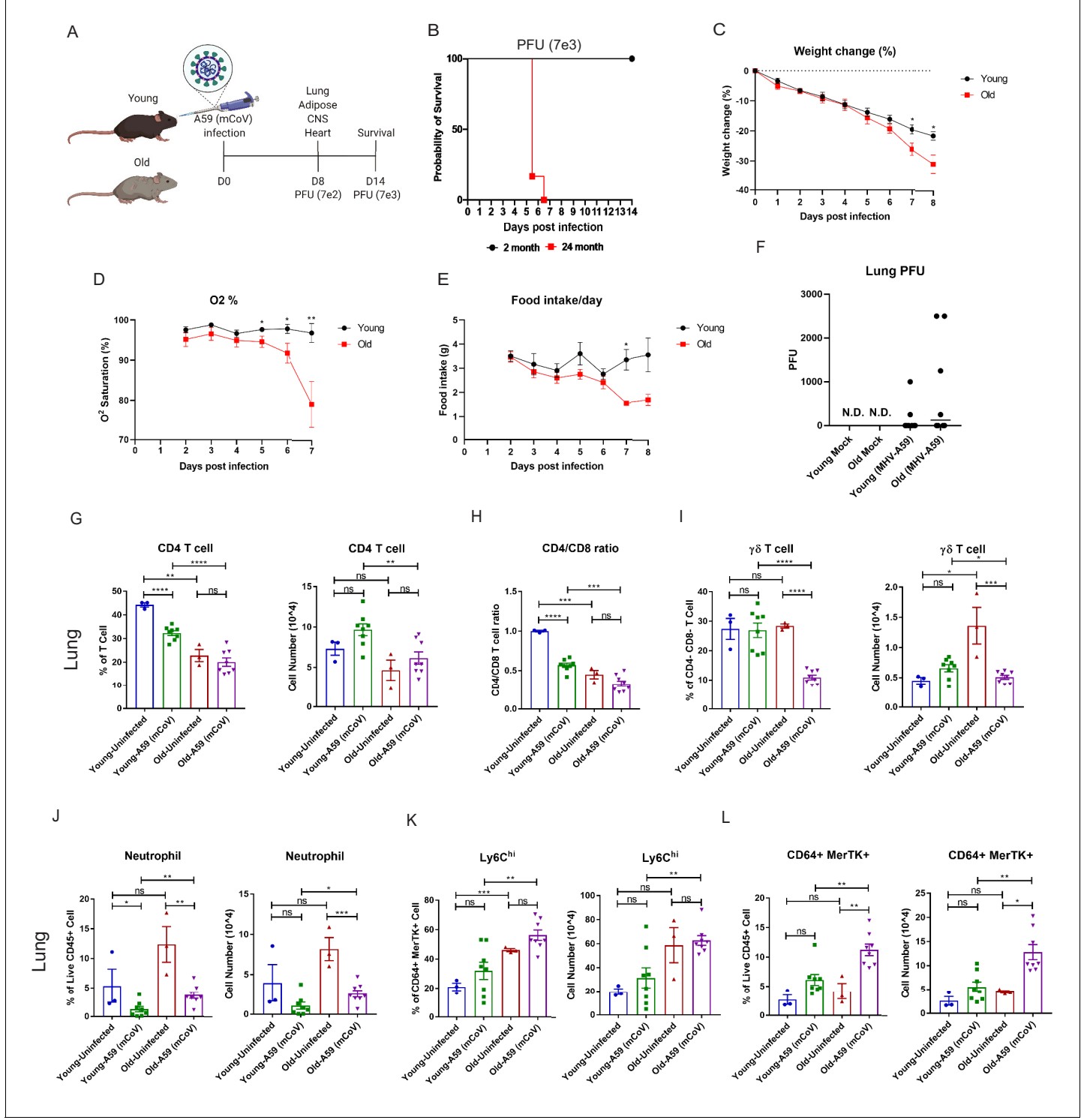

**Figure 1.** Aging exacerbates A59 (mCoV) infection. (**A**) Schematic of A59 (mCoV) infection experiment with young (2–6 months) and old mice (20–24 months). (**B**) Probability of survival of young (n = 6) and old (n = 6) infected mice. Survival of mice was examined after infection with high dose of virus (PFU 7e3) up to 14 days. (**C–E**) Young (n = 8) and old mice (n = 8) were infected with low dose of virus (PFU 7e2), and weight change (%) (**C**), $O_2$ saturation (**D**), and daily food intake (**E**) were recorded. (**F**) Plaque assay of lung from uninfected, infected young and old mice. (**G–L**) Flow cytometry analysis of CD4 T cell (**G**), CD4/CD8 T cell ratio (**H**), γδ T cell (**I**), neutrophil (**J**), Ly6C$^{hi}$ cell (**K**), and CD64$^+$ MerTK$^+$ cell (**L**) on day 8 post (PFU 7e2) infection. Error bars represent the mean ± S.E.M. Two-tailed unpaired t-tests were performed for statistical analysis. *p<0.05; **p<0.01; ***p<0.001; ****p<0.0001.

The online version of this article includes the following figure supplement(s) for figure 1:

*Figure 1 continued on next page*

**Figure supplement 1.** Characterization of immune cell population in young and old mice infected with A59 (mCoV).

months) (*Figure 1A*). The LD-0 infectious dose of mCoV-A59 in adult (PFU 7e3) caused 100% lethality in aged mice (*Figure 1B*). Aged mice displayed greater weight loss (*Figure 1C*), hypoxemia (*Figure 1D*), and anorexia (*Figure 1E*), without a significant difference in viral load in lungs (*Figure 1F*) after infection (PFU 7e2) when compared to adults.

Immune profiling of the lung revealed that aged mice had significant reductions both in the percentage of CD4 cells and in the CD4:CD8 ratio in the lung compared to adult controls at steady state and after infection (*Figure 1G,H*, *Figure 1—figure supplement 1A*). Interestingly, while $\Upsilon\delta$ T cell numbers in the lung were significantly increased at steady state in aged mice compared to adult mice, both the proportion and number of $\Upsilon\delta$ T cells in the lung were dramatically reduced only in aged mice after infection compared to adult mice (*Figure 1I*, *Figure 1—figure supplement 1A*). In the spleen, $\Upsilon\delta$ T cells were found to be reduced at steady state in aged mice (*Figure 1—figure supplement 1B*).

Analyses of the myeloid compartment revealed that aged mice had significantly increased neutrophils in the lung post-infection with a trend toward increase in neutrophils at steady state (*Figure 1J*). We did not detect age-associated changes in eosinophils in lungs (*Figure 1—figure supplement 1C*). Aging increased Ly6C$^{hi}$ monocytes, which rose further post-infection (*Figure 1K*, *Figure 1—figure supplement 1D*). Moreover, the frequency and number of CD64$^+$MerTK$^+$ cells were highest in infected aged mice (*Figure 1L*, *Figure 1—figure supplement 1D*), while no significant differences were observed in the total population of alveolar or interstitial macrophages (*Figure 1—figure supplement 1E,F*) in lungs. These data suggest that post-sub-lethal infection, when young animals recover, aging is associated with delayed resolution of infection with prolonged infiltration of inflammatory myeloid cells and a dramatic reduction in γδ T cells in the lung.

Transmission electron microscopy confirmed the dissemination of the viral particles in pneumocytes in lungs (*Figure 2A*). The pneumotropism of mCoV-A59 was also validated by demonstration of viable mCoV-A59 in lungs using plaque assays, and aged mice did not show significant differences in viral load in lungs (*Figure 1F*). Interestingly, following mCoV-A59 inoculation, pathology analyses by hematoxylin and eosin (H and E) and MSB staining, in both 6 month and 20–24 month old mice, revealed perivascular inflammation (arrows, arrowhead) as well as perivascular edema (*) and increased perivascular collagen/fibrosis (MSB, blue) that is more severe in the 20–24 month mCoV-A59 infected mice (*Figure 2B*). Furthermore, 20–24 month old mice intranasally inoculated with mCoV-A59 have dense foci visible at low power (box) and amphophilic material (fibrosis) with few scattered brightly eosinophilic erythrocytes (gray arrowhead) admixed with lymphocytes and plasma cells. By MSB stain, at higher power, this same focus (***) in the 20–24 month infected mice revealed that the end of a small blood vessel (BV) terminates into a mass of collapsed alveoli, without obvious septa admixed with inflammatory cells, disorganized fibrin/collagen fibers (blue) suggestive of ante mortem pulmonary thrombosis in contrast with post-mortem blood clots where erythrocytes are yellow (** MSB, yellow) (*Figure 2B*). Taken together, consistent with ARDS, the lungs of aged mice infected with mCoV-A59 had increased foci of inflammation, immune cell infiltration, perivascular edema, hyaline membrane formation and type II pneumocyte hyperplasia, organizing pneumonia, interstitial pneumonitis, and occasional hemorrhage and microthrombi, affecting approximately 75% of the lungs (*Figure 2B*).

## Aging enhances systemic inflammatory response in mCoV-A59 infected mice

We next investigated whether mCoV-A59 infection in aged mice mimics the hyperinflammatory systemic response seen in elderly patients infected with COVID-19. Compared to young animals, old mice infected with equivalent doses of mCoV-A59 displayed significant increase in circulating IL-1β, TNFα, IL-6, and MCP-1 (*Figure 3A–C*, *Figure 3—figure supplement 1A–D*), without affecting MIP-1β (*Figure 3—figure supplement 1E*). Similar to COVID-19, infection with mCoV-A59 caused increased cardiac inflammation in old mice as evaluated by greater number of infiltrating CD68$^+$ myeloid cells (*Figure 3D*, *Figure 3—figure supplement 1F*).

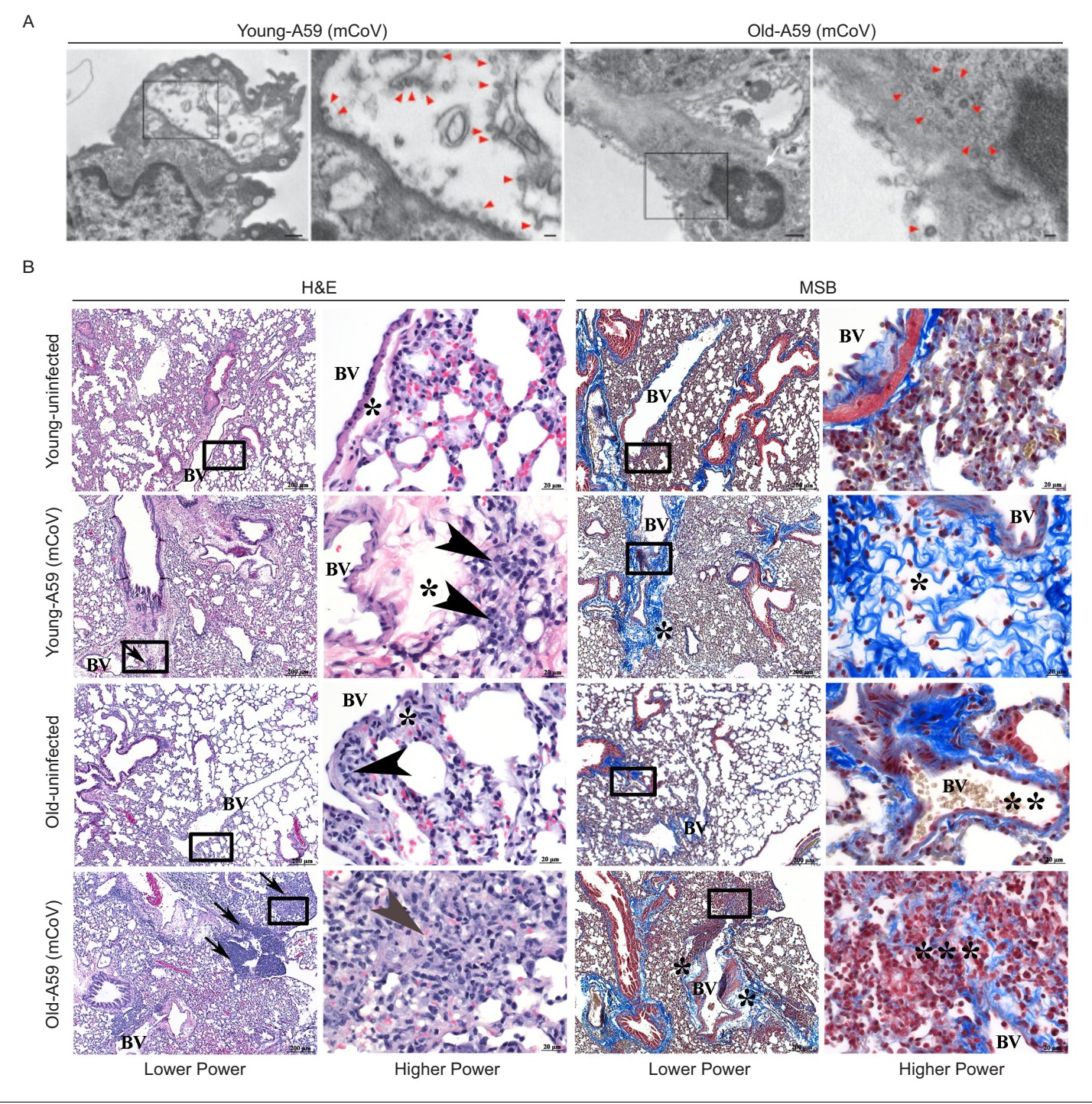

**Figure 2.** A59 (mCoV) infection significantly affects lung phenotype in old mice. (**A**) Transmission electron microscopic images of A59 (mCoV) particles in pneumocytes. Left panels show pneumocyte of an infected young (left) and old (right) mice. Bar scale represents 500 nm. Right panels show zoomed-in images of boxed areas of left panels. Pneumocyte with budding viral particles was indicated by red arrowheads. Bar scale represents 100 nm. Apoptotic pneumocyte of an infected old mouse showed the shrunken and degraded nucleus (white arrow) and chromatin condensation (white asterisk). (**B**) Representative photomicrographs of hematoxylin and eosin (H and E)-stained (left) and Martius scarlet blue trichrome (MSB)-stained (right) sections of lung from young and old mice 8 days post infection with A59 (mCoV), along with lung from uninfected young and old mice. There are foci of inflammation (arrow), perivascular edema (*), and perivascular lymphocytes and plasma cells (arrow head). Boxed areas in low power images were used for high power imaging. BV indicates small blood vessel.

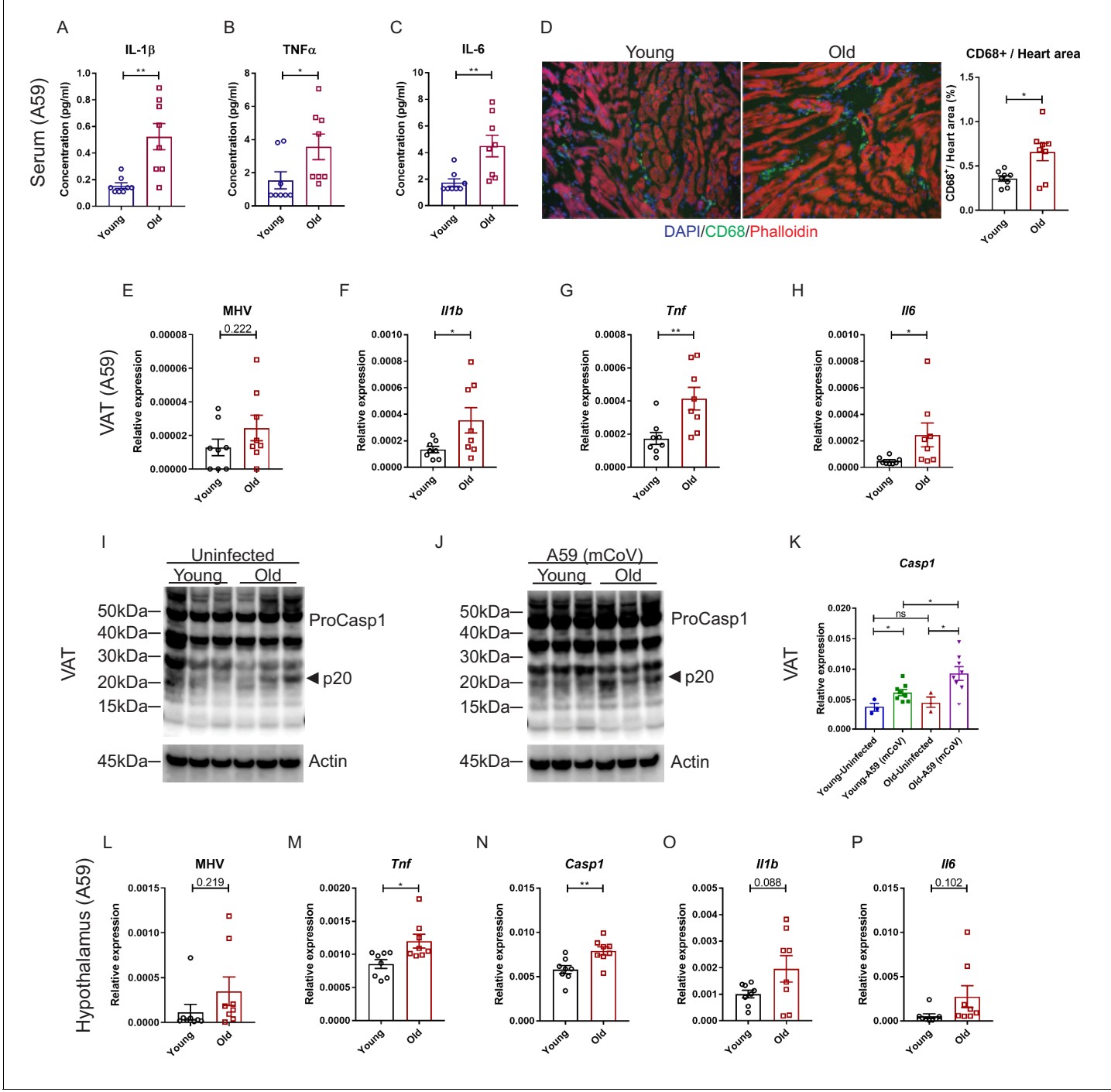

**Figure 3.** Aging induces systemic hyperinflammatory response in A59 (mCoV) infection. (A–C) Serum levels of inflammatory cytokines IL-1β (A), TNFα (B), and IL-6 (C) of young (6 months old) and old (24 months old) infected mice on day 8 post (PFU 7e2) infection. (D) Representative immunofluorescence analysis of CD68 expression, phalloidin, and DAPI in hearts isolated from young and old mice. CD68+ cells/heart area analysis was shown in right panel. (E) Quantification of MHV-A59 (mCoV) in visceral adipose tissue (VAT) of young and old infected mice by quantitative PCR (qPCR) of a gene expressing A59 M protein. (F–H) qPCR analysis of *Il1b* (F), *Tnf* (G), and *Il6* (H) in VAT of young and old infected mice. (I) Western blot analysis of caspase-1 inflammasome activation in VAT of uninfected young and old mice. (J) Immunoblot analysis of caspase-1 cleavage showing higher inflammasome activation in VAT in aged mice post-infection. (K) Quantification of gene expression of *Casp1* in VAT by qPCR. (L) Quantification of MHV-A59 (mCoV) in hypothalamus of young and old infected mice by qPCR. (M–P) Gene expression analysis of *Tnf* (M), *Casp1* (N), *Il1b* (O), and *Il6* (P) in hypothalamus of young and old mice 8 days post-infection. Error bars represent the mean ± S.E.M. Two-tailed unpaired t-tests were performed for statistical analysis. *p<0.05; **p<0.01.

*Figure 3 continued on next page*

*Figure 3 continued*

The online version of this article includes the following figure supplement(s) for figure 3:

**Figure supplement 1.** Inflammatory response in young and old mice infected with A59 (mCoV).

Given that increased visceral adiposity is a risk factor for COVID-19 severity and expression of ACE2 is upregulated in adipocytes of obese and diabetic patients infected with SARS-CoV-2 (*Kruglikov and Scherer, 2020*), we next studied whether mCoV-A59 infection affects adipose tissue. Given the prevalence of obesity is 10% among younger adults aged 20–39, 45% among adults aged 40–59 years, and 43% among older adults aged 60 and over (*Hales et al., 2020*), we investigated adipose tissue inflammation as a potential mechanism that contributes to infection severity in the aged. Interestingly, consistent with the prior findings that adipose tissue can harbor several viruses (*Damouche et al., 2015*), the mCoV-A59 RNA was detectable in visceral adipose tissue (VAT) (*Figure 3E*, *Figure 3—figure supplement 1G*). Despite similar viral loads, VAT of aged infected mice had significantly higher mRNA levels of the pro-inflammatory cytokines IL-1β, TNFα, and IL-6 (*Figure 3F–H*, *Figure 3—figure supplement 1H–J*). Moreover, consistent with prior work, aging is associated with increases in inflammasome activation (*Figure 3I*). Infection with mCoV-A59 further enhanced caspase-1 cleavage (p20 active heterodimer) as well as expression of inflammasome components, *Casp1* and *Nlrp3*, in old mice (*Figure 3I–K*, *Figure 3—figure supplement 1K*). In addition, similar to SARS-CoV-2 invasiveness in central nervous system (CNS), the mCoV-A59 was detectable in the hypothalamus (*Figure 3L*, *Figure 3—figure supplement 1L*). Compared to adults, the hypothalamus of aged infected mice showed increased mRNA expression of TNFα and caspase-1 (*Figure 3M,N*, *Figure 3—figure supplement 1M,N*) with no significant differences in IL-1β, IL-6 (*Figure 3O,P*, *Figure 3—figure supplement 1O,P*), and NLRP3 (*Figure 3—figure supplement 1Q*). Infection in both young and aged mice caused significant increases in markers of astrogliosis and microglia activation (*Figure 3—figure supplement 1R,S*). Interestingly, mCoV-A59 reduced the mRNA expression of orexigenic neuropeptide-Y (*Figure 3—figure supplement 1R,S*), consistent with the fact that infected mice display anorexia (*Figure 1C,E*). However, mCoV-A59 infection completely abolished the expression of pre-opiomelanocortin (POMC) in the hypothalamus, a transcript expressed by POMC neurons, which is involved in the control of the autonomic nervous system and integrative physiology. Further investigation will be necessary to test the involvement of the hypothalamus in the pathogenesis of COVID-19 and organ failure due to alterations in the autonomic nervous system.

## Ketogenic diet-mediated protection against mCoV-A59 infection in aging is coupled with inflammasome deactivation

Given the switch from glycolysis to fatty acid oxidation reprograms the myeloid cell from pro-inflammatory to tissue reparative phenotype during infections (*Ayres, 2020*; *Buck et al., 2017*; *Galván-Peña and O'Neill, 2014*), we next investigated whether mCoV-A59-driven hyperinflammatory response in aging can be targeted through immunometabolic approaches. Hepatic ketogenesis, a process downstream of lipolysis that converts long-chain fatty acids into short chain β-hydroxybutyrate (BHB) as a preferential fatty acid fuel during starvation or glucoprivic states, inhibits the NLRP3 inflammasome activation (*Youm et al., 2015*) and protects against influenza infection-induced mortality in mice (*Goldberg et al., 2019*). We infected bone marrow-derived macrophages (BMDMs) with mCoV-A59 *in vitro* in TLR4 (*Figure 4A*, *Figure 4—figure supplement 1A*) and TLR1/2 primed cells (*Figure 4B*, *Figure 4—figure supplement 1B*). Infection with mCoV-A59 caused robust activation of inflammasome as measured by cleavage of active IL-1β (p17) in BMDM supernatants (*Figure 4A,B*) as well as in cell lysates (*Figure 4—figure supplement 1A,B*). Given our prior findings that ketone metabolites specifically inhibit the NLRP3 inflammasome in response to sterile DAMPs such as ATP, ceramides, silica, and urate crystals (*Youm et al., 2015*), we next tested whether BHB impacts inflammasome activation caused by mCoV-A59.

Interestingly, BHB treatment reduced pro and active cleaved IL-1β (p17) in both conditions when protein level was measured in the supernatant (*Figure 4A,B*) and cell lysate (*Figure 4—figure supplement 1A,B*). Mechanistically, post-mCoV-A59 infection, the BHB reduced the oligomerization of ASC, which is an adaptor protein required for the assembly of the inflammasome complex

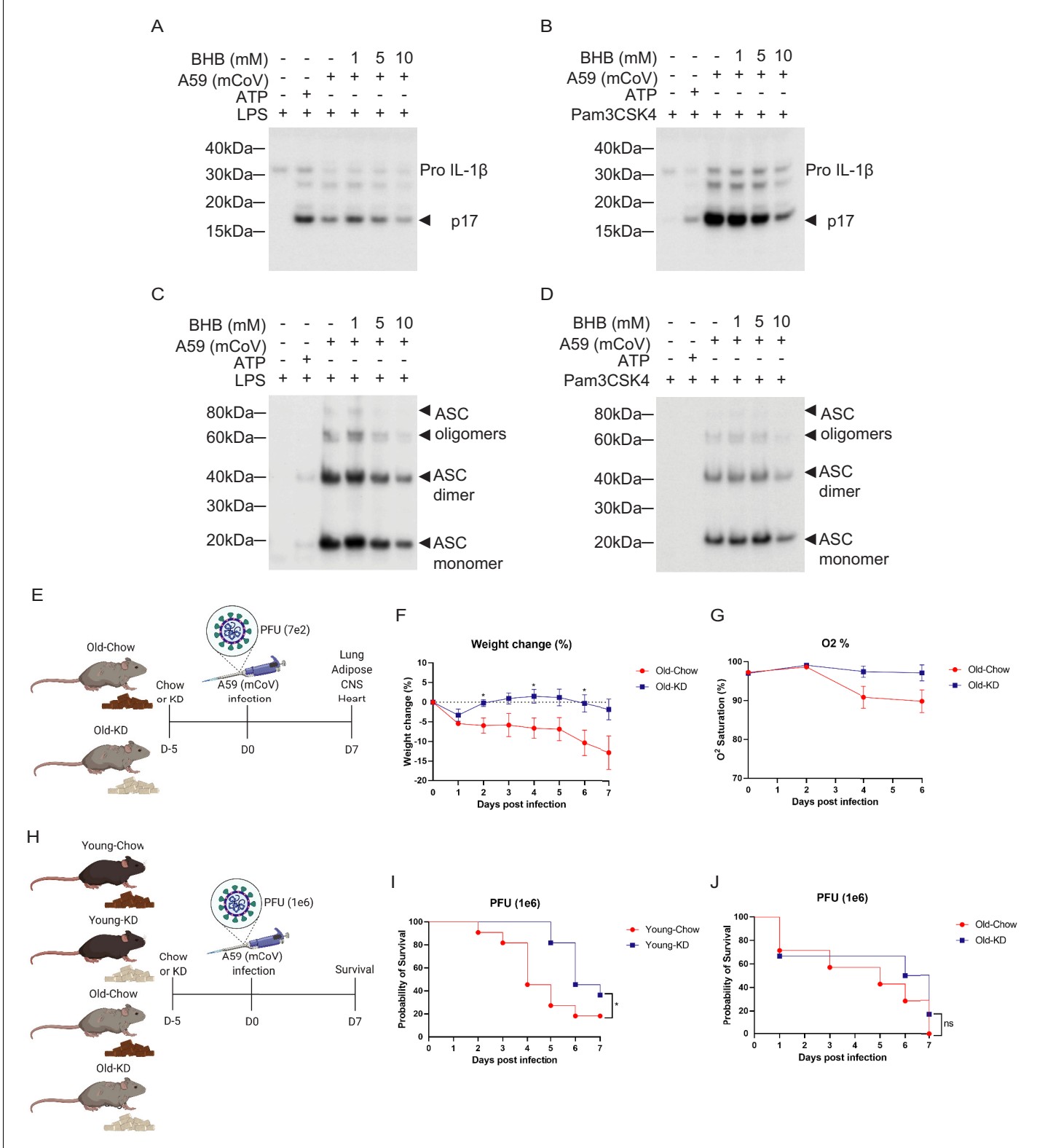

**Figure 4.** Ketogenic diet reduces the severity of A59 (mCoV) infection in old mice. (**A, B**) Western blot analysis about pro and active cleaved p17 form of IL-1β from supernatant of A59 (mCoV)-infected BMDMs co-treated with priming reagents such as LPS (**A**) and Pam3CSK4 (**B**), and BHB with indicated concentration. (**C, D**) Western blot analysis of ASC monomer, dimer, and oligomers from insoluble pellet of A59 (mCoV) infected BMDMs co-treated with priming reagents such as LPS (**C**) and Pam3CSK4 (**D**), and BHB with indicated concentration. (**E**) Schematic of non-lethal dose (PFU 7e2) of A59

*Figure 4 continued on next page*

Figure 4 continued

(mCoV) infection experiment with old mice (20–21 month) fed chow (Old-Chow, n = 6) or ketogenic diet (Old-KD, n = 5). The mice were provided with diet from 5 days before infection. (F, G) After infection, the phenotype was evaluated until 7 days post-infection. Weight change (%) (F), and % $O_2$ saturation (G) in old mice fed chow or KD. (H) Schematic of lethal dose (PFU 1e6) of A59 (mCov) infection experiment with young (4 months) and old (24 months) mice fed chow (Young-Chow, n = 11; Old-Chow, n = 7) or KD (Young-KD, n = 11; Old-KD, n = 6). The mice were provided with diet from 5 days before infection. After infection, the survival was evaluated until 7 days post-infection. (I) Probability of survival of lethal dose A59 (mCoV)-infected young mice fed chow or KD. (J) Probability of survival of lethal dose A59 (mCoV)-infected old mice fed chow or KD. Error bars represent the mean ± S. E.M. Two-tailed unpaired t-tests were performed for statistical analysis. Gehan–Breslow–Wilcoxon tests were performed for survival analysis. *p<0.05. The online version of this article includes the following figure supplement(s) for figure 4:

**Figure supplement 1.** Protective effect of BHB in BMDM against A59 (mCoV) and phenotype of A59 (mCoV) infected old mice fed chow or KD.

(*Figure 4C,D*). This data provides evidence that the ketone metabolite BHB can lower inflammation in response to coronavirus infection and deactivate the inflammasome. However, inflammasome activation is also required for mounting adequate immune response against pathogens including certain viruses. Therefore, we next investigated if induction of ketogenesis and ketolysis *in vivo* by feeding a diet rich in fat and low in carbohydrates that elevates BHB level impacts inflammasome and host defense against mCoV-A59 infection in aged mice.

We next tested whether ketogenic diet that increases BHB via mitochondrial metabolism affects the outcome of coronavirus infection in mice. Ketogenesis is dependent on hydrolysis of triglycerides and conversion of long-chain fatty acids in liver into short chain fatty acid BHB that serves as primary source of ATP for heart and brain when glucose is limiting. Aging is associated with impaired lipid metabolism which includes reduced lipolysis that generates free fatty acids that are essential substrates for BHB production. Thus, it is unclear whether in context of severe infection and aging, if sufficient ketogenesis can be induced. To test this, the aged male mice (20–21 months old) were fed a ketogenic diet (KD) or control diet for 5 days and then intranasally infected with mCoV-A59 (*Figure 4E*). Despite mCoV-A59's known effects in causing hepatic inflammation (*Navas et al., 2001*), we observed that compared to chow-fed animals, old KD-fed mice achieved mild physiological ketosis between 0.6 and 1 mM over the course of infection for 1 week (*Figure 4—figure supplement 1C*).

Compared to chow-fed animals, the mCoV-A59-infected mice fed KD displayed similar levels of food intake, blood glucose, core body temperature, heart rate, and respiration (*Figure 4—figure supplement 1D–I*). Interestingly, mCoV-A59-infected KD-fed mice were protected from infection-induced weight loss and hypoxemia (*Figure 4F,G*). Furthermore, we infected young and old mice fed either chow or KD with a lethal dose of mCoV-A59 (PFU 1e6) to investigate whether KD affects mortality caused by mCoV-A59 infection (*Figure 4H*). Interestingly, compared to chow-fed controls, young mice fed KD prior to infection showed improved survival (*Figure 4I*). However, old mice fed KD were not protected from death caused by high-dose mCoV-A59-induced infection (*Figure 4J*). Together, these data show that KD is protective in aged mice in sub-lethal infections and partially protects adult, but not aged animals, from lethal coronavirus infections.

### Induction of ketogenic substrate switch inhibits systemic inflammation in aged mCoV-A59 infection

Elderly COVID-19 patients exhibit multi-organ failure with systemic viremia and inflammation. Therefore, we next investigated the impact of KD on the inflammatory response in lungs, adipose tissue, and hypothalamus in old mice post-sub-lethal mCoV-A59 infection. Consistent with the improved clinical outcome and protection afforded by ketone bodies in infected mice, we found that KD-fed mice inoculated with mCoV-A59 had significantly reduced mRNA expression of pro-inflammatory cytokines IL-1β, TNFα, and IL-6 in lung, VAT, and hypothalamus (*Figure 5A–C*). Aging and mCoV-A59 increases inflammasome activation, which is increasingly implicated in the pathogenesis of COVID-19 (*Vijay et al., 2017*; *Youm et al., 2013*). The SARS-CoV open-reading frame 3a (ORF3a) and ORF8b activates the NLRP3 inflammasome (*Siu et al., 2019*) by inducing ER stress and lysosomal damage (*Shi et al., 2019*). Moreover, ability of bats to harbor multiple viruses including coronaviruses is due to splice variants in the LRR domain of NLRP3, which prevents inflammasome-mediated inflammatory damage (*Ahn et al., 2019*). Interestingly, in the aging mouse model of mCoV-A59 infection, KD significantly lowered NLRP3 and caspase-1 mRNA in lung, VAT, and

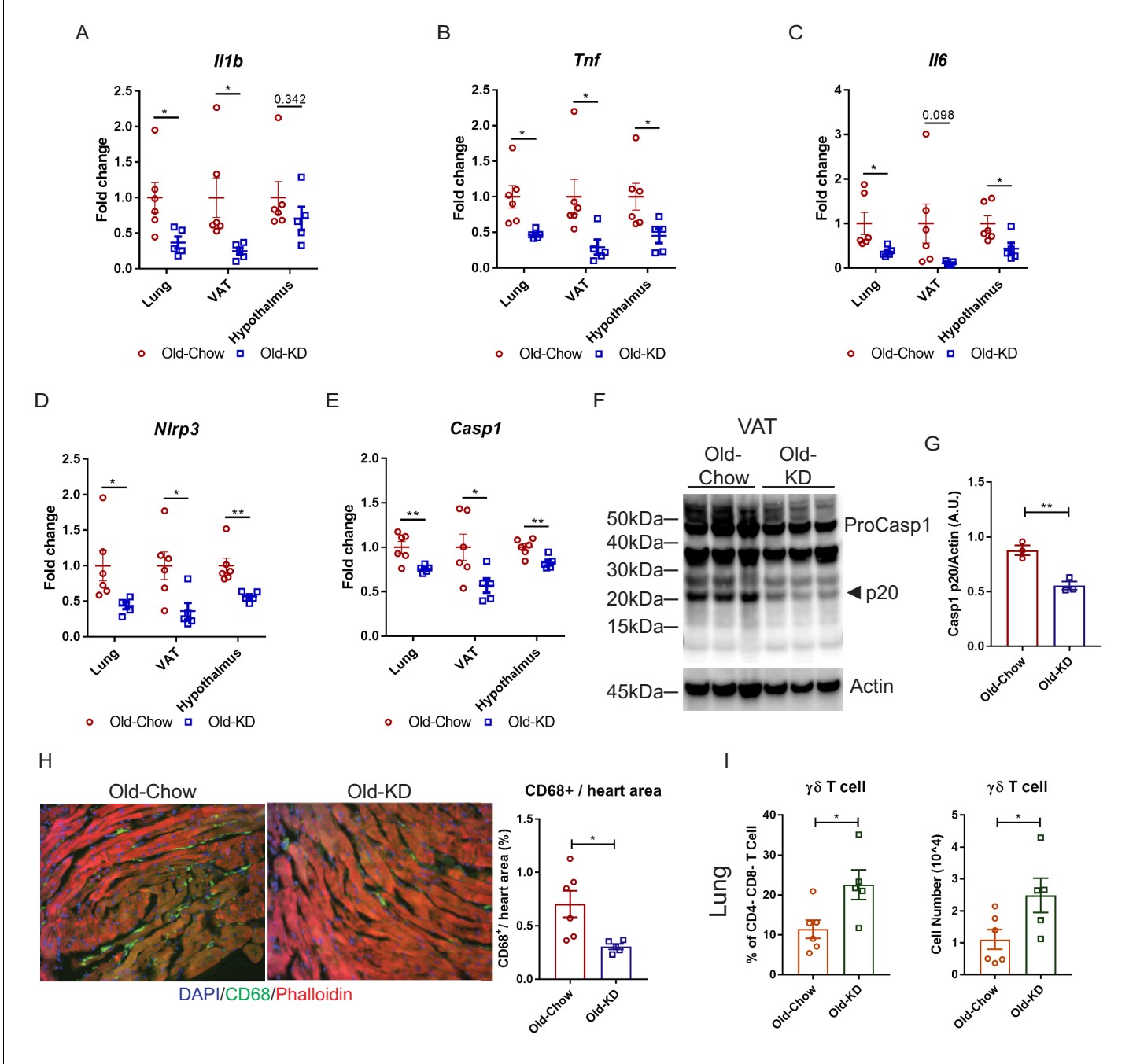

**Figure 5.** Ketogenic diet protects old mice from A59 (mCoV) infection by alleviation of inflammation. (A–E) Gene expression analysis of inflammatory cytokines (A–C) and components of inflammasome (D, E) in lung, VAT, and hypothalamus of A59 (mCoV)-infected old mice fed chow (Old-Chow, 20–21 months, n = 6) or KD (Old-KD, 20–21 months, n = 5). (F, G) Western blot analysis of caspase-1 inflammasome activation (F) in VAT of infected Old-Chow and Old-KD mice with quantification (G). (H) Representative immunofluorescence images of CD68 expression, phalloidin, and DAPI in hearts isolated from Old-Chow and Old-KD mice (left) and CD68+ cells/heart area analysis in heart of infected Old-Chow and Old-KD mice (right). (I) Flow cytometry analysis of γδ T cell in lung of infected Old-Chow and Old-KD mice. Error bars represent the mean ± S.E.M. Two-tailed unpaired t-tests were performed for statistical analysis. *p<0.05; **p<0.01.

The online version of this article includes the following figure supplement(s) for figure 5:

**Figure supplement 1.** Immune cell population profile in lung of infected old mice provided with chow or ketogenic diet.

**Figure supplement 2.** Myeloid cell population profile in lung of infected old mice provided with chow or ketogenic diet.

hypothalamus (*Figure 5D,E*), inflammasome activation in VAT (*Figure 5F,G*), and decreased myeloid cell infiltration in heart (*Figure 5H*). The ketogenesis in infected old mice did not affect the frequency of CD4, CD8 effector memory or macrophage subsets in lungs, suggesting that reduction in pro-inflammatory cytokines was not a reflection of reduced infiltration of these cell types (*Figure 5—figure supplements 1A–L* and *2A–H*). Interestingly, we found that KD feeding rescued mCoV-A59-induced depletion of γδ T cell in lungs of aged mice (*Figure 5I*, *Figure 5—figure supplement 1A*).

## Ketogenesis induces protective γδ T cells and decreases myeloid cell subset in mCoV-A59-infected old mice

To determine the mechanism of ketogenesis-induced protection from mCoV-A59-driven inflammatory damage in aging, we next investigated the transcriptional changes in lung at the single-cell level. The scRNA-sequencing of whole-lung tissues (*Figure 6A*, *Figure 6—figure supplement 2A*) found that KD feeding in old infected mice caused significant increase in goblet cells (*Figure 6B*), expansion of γδ T cells (*Figure 6B*) and significant decrease in proliferative cell subsets and monocyte populations (*Figure 6B*). When differential gene expression analyses were performed with the clusters, we observed only very modest changes within few clusters and no shared signature (*Figure 6—figure supplement 2B*). Comparison with scRNA-seq of the lungs from young and old infected animals highlighted the largest increase in B cells and club cells and reduction of proliferating myeloid cells, including Trem2+ macrophages and NK cells (*Figure 6—figure supplement 1A–D*). The fraction of Foxp3-positive Treg cells were not different between young infected and old infected mice (*Figure 6—figure supplement 1E–G*). Notably, when differential gene expression and pathway analyses were performed (*Figure 6—figure supplement 1H–K*), the old infected mice showed a reduced interferon signature, suggesting increased vulnerability to the viral infection (*Figure 6—figure supplement 1I,J*). Interestingly, some of the most striking changes occurred in T cells, where ketogenesis led to a substantial increase in γδ but not αβ T cells (*Figure 6C*, *Figure 6—figure supplement 2C*). To understand whether expansion of γδ T cells was also accompanied by the changes in their regulatory programs, we sorted the lung γδ T cells from aged mice fed chow diet and KD and conducted bulk RNA-sequencing to determine the mechanism of potential tissue protective effects of these cells in mCoV-A59 infection. We found that KD in aging significantly increased the genes associated with reduced inflammation (*Figure 6D*), increased lipoprotein remodeling and downregulation of TLR signaling, Plk1 and aurora B signaling pathways in γδ T cells (*Figure 6E*). However, there was no difference in mRNA expression levels of Vγ chain subtypes, IL-17, or IFN-γ (*Figure 6—figure supplement 2D–F*). Furthermore, RNA-sequencing revealed that lung γδ T cells from ketogenic mCoV-A59-infected old mice displayed elevated respiratory electron transport and complex I biogenesis (*Figure 6E*). In addition, Golgi to ER retrograde transport and cell cycle are downregulated, suggesting the reduced activation status of γδ T cell (*Figure 6E*). These data suggest that γδ T cells expanded with KD are functionally more homeostatic and immune protective against mCoV-A59 infection.

Zooming in into monocyte sub-population, we observed three distinct monocyte clusters (*Figure 6F*), characterized by *Ifi44*, *Lmna*, and *Cd300e*, respectively (*Figure 6G*). Strikingly, ketogenesis-induced change in the monocyte compartment was driven by a loss of cluster 1 (characterized by high levels of *Chil3*, *Lmna*, *Il1r2*, *Lcn2*, *Cd33*, *Cd24a*, *Figure 6H, Figure 6—figure supplement 2G,H*). The same clusters were identified in scRNA-sequencing with young and old infected mice and aging was only associated with reduction of the cluster 0 (*Figure 6—figure supplement 2I–K*). In addition, the loss of monocyte sub-population was observed in cells with low interferon response, further suggesting the immune protective response induction post-ketogenesis in infected mice. This is an intriguing finding that is consistent with recent observations that dietary interventions can impact plasticity of the monocyte pool in both mouse and human (*Collins et al., 2019*; *Jordan et al., 2019*).

## Discussion

Immune-senescence exemplified by inflammasome-mediated basal activation of myeloid cells, expansion of pro-inflammatory aged B cells, impaired germinal center, and antibody responses together with thymic demise and restriction of T cell repertoire diversity all contribute to increased risk of infections and vaccination failures in elderly (*Akbar and Gilroy, 2020*; *Frasca et al., 2020*;

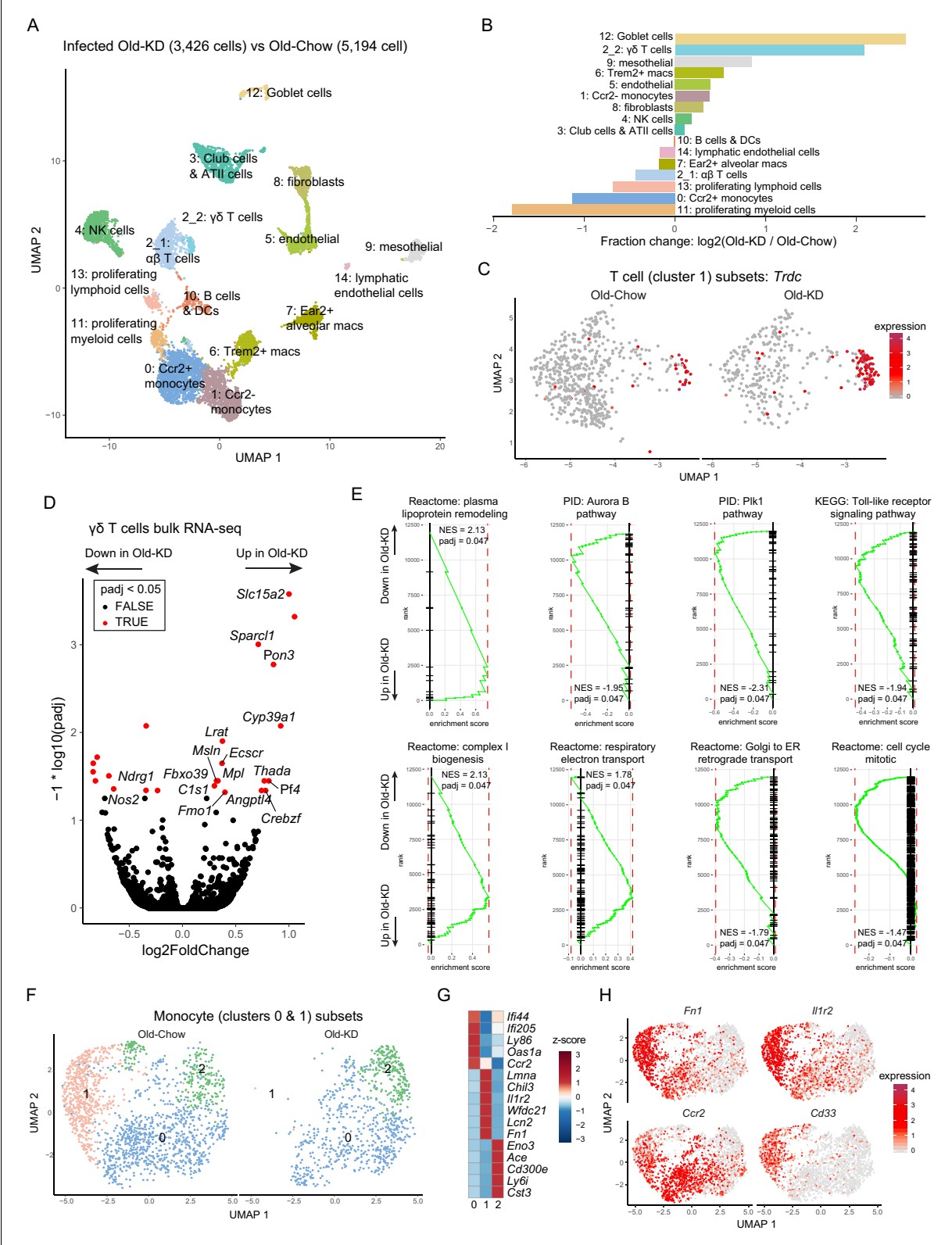

**Figure 6.** Ketogenesis induces protective γδ T cell expansion and inhibits myeloid cell activation in aged mice with mCoV infection. (**A**) Uniform Manifold Approximation and Projection (UMAP) plot of lung cells from Old-KD and Old-Chow samples as described in *Figure 4E*. (**B**) Bar chart shows population fold-changes in relative abundance of each cluster. (**C**) Zoom into UMAP plot from A showing T cells (cluster 1) split by sample. Color represents expression of *Trdc*. (**D**) Volcano plot identifying significantly regulated genes (5% FDR) within sorted γσ T cells from lungs of Old-KD and

*Figure 6 continued on next page*

Figure 6 continued

Old-Chow mice. Increase in expression corresponds to ketogenic diet-induced genes. (E) GSEA enrichment curve based on differential expression analysis results shown in (D). (F) Monocyte cluster 0 and 1 from (A) were subset and analyzed separately. UMAP plot of monocytes split by samples. (G) Heatmap of normalized within row gene expression values of selected markers of three monocyte subsets. (H) UMAP as in (F). Color represents expression of selected genes. For (A–C, F–H), expression values were obtained by pooling data from Old-Chow and Old-KD samples (each containing n = 6 chow and n = 5 KD pooled biological samples into one technical sample for each diet).

The online version of this article includes the following figure supplement(s) for figure 6:

**Figure supplement 1.** Single-cell RNA-sequencing analysis of lung from young and old A59 (mCoV)-infected mice.
**Figure supplement 2.** The lung RNA-sequencing analysis of old infected mice fed chow or ketogenic diet.

*Goldberg and Dixit, 2015*; *Goronzy and Weyand, 2019*). It is likely that multiple mechanisms partake in aging-induced mortality and morbidity to SARS-CoV-2. However, study of immunometabolic mechanisms that control aberrant inflammatory response in elderly COVID-19 patients is incompletely understood. Epidemiological data strongly support that elderly and aged individuals with late-onset chronic diseases – including diabetes, obesity, heart conditions, pulmonary dysfunctions, and cancer – present a much higher disease severity compared to young healthy adults (*Cai et al., 2020*; *Chen et al., 2020*). These observations suggest that it is the vulnerability of the various tissues that occur in these chronic conditions that predispose elderly to develop severe forms of COVID-19.

Rodent CoVs are natural, highly contagious pathogens of mice and rats (*Compton et al., 2003*; *Compton et al., 1993*). These viruses also offer the possibility of efficient and safe platforms for modeling COVID-19 and examining immunometabolic mechanisms and interventions that impact disease. Of particular interest for COVID-19 are the strains of MHV that are respiratory tropic (*Yang et al., 2014*). Given, these advantages, MHV mCoV-A59 infection in C57BL/6 mice can be a powerful tool to rapidly study the disease as well as test therapeutic interventions. We demonstrate that MHV mCoV-A59 infection in aged mice recapitulates severe features of COVID-19 that includes up to 30% weight loss, sickness behavior exemplified by anorexia, loss of oxygen saturation, lung pathology including neutrophilia, monocytosis, loss of γδ T cells, lymphopenia, increase in circulating pro-inflammatory cytokines, hypothalamic, adipose and cardiac inflammation, and inflammasome activation. Importantly, LD-0 dose of MHV mCoV-A59 induces 100% lethality in 2 year old male mice, suggesting that this model allows investigation of COVID-19-relevant immunometabolic mechanisms that control disease development and severity with aging.

Mechanistically, NLRP3 inflammasome has been demonstrated to be an important driver of aging-induced chronic inflammation and organ damage (*Bauernfeind et al., 2016*; *Camell et al., 2017*; *Youm et al., 2013*). COVID-19 patients have inflammasome-dependent pyroptosis and increase in IL-18 (*Lucas et al., 2020*; *Zhou et al., 2020*). Consistent with the hypothesis that aging may exacerbate inflammasome activation in SARS-CoV-2 infection, our data demonstrates that *in vivo*, mCoV infection increases NLRP3 inflammasome-mediated inflammation. Recent study shows that MHV-A59 also activates the NLRP3 inflammasome *in vitro* BMDM (*Zheng et al., 2020*). In addition, severe cases of COVID-19 are accompanied with dysregulation of monocyte populations with increased level of S100A8/A9 or calprotectin (*Schulte-Schrepping et al., 2020*; *Silvin et al., 2020*), which can prime and induce the inflammasome activation (*Goldberg et al., 2017*). Interestingly, the KD feeding blocked infiltration of pathogenic monocyte subset in lungs that has high S100A8/9 and low interferon expression. These data underscore that enhanced innate immune tolerance mediated by inflammasome deactivation maybe an important strategy against COVID-19.

The integrated immunometabolic response (IIMR) is critical in regulating the setpoint of protective versus pathogenic inflammatory response (*Lee and Dixit, 2020*). The IIMR involves sensing of nutrient balance by neuronal (sympathetic and sensory innervation) and humoral signals (e.g. hormones and cytokines) between the CNS and peripheral tissues that allow the host to prioritize storage and/or utilize substrates for tissue growth, maintenance, and protective inflammatory responses. Peripheral immune cells, both in circulation and those residing within tissues, are subject to regulation by the metabolic status of the host. Ketone bodies, BHB, and acetoacetate are produced during starvation to support the survival of host by serving as an alternative energy substrate when glycogen reserves are depleted (*Newman and Verdin, 2017*). Classically, ketone bodies are considered essential metabolic fuels for key tissues such as the brain and heart (*Puchalska and Crawford, 2017*; *Veech et al., 2017*). However, there is increasing evidence that immune cells can also be profoundly

regulated by ketone bodies (*Goldberg et al., 2020*; *Youm et al., 2015*). For example, stable isotope tracing revealed that macrophage oxidation of liver-derived AcAc was essential for protection against liver fibrosis (*Puchalska et al., 2019*). Given our past findings that ketone bodies inhibit NLRP3 inflammasome activation induced by sterile DAMPs, we next hypothesized that coronavirus mediated inflammasome activation and disease severity in aging could be improved by BHB driven improved metabolic efficiency and NLRP3 deactivation. In support of this hypothesis, we found that BHB inhibits the mCoV-A59-induced NLRP3 inflammasome assembly and KD reduces caspase-1 cleavage as well as decreases gene expression of inflammasome components. We next investigated the mechanism of protection elicited by KD that is relevant to aging. Interestingly, scRNA-sequencing analyses of lung homogenates of old mice fed KD revealed robust expansion of immunoprotective γδ T cells, which are reported to decline in COVID-19 patients (*Lei et al., 2020*; *Rijkers et al., 2020*). The KD activated the mitochondrial function as evidenced by enhanced complex-1 biogenesis and upregulation of ETC in immunoprotective γδ T cells. These results are consistent with a previous study reporting that KD-mediated enhancement of mitochondrial fatty acid oxidation is required for protection against flu infection. In the study, increasing BHB by feeding ketone esters could not mimic the effect of KD (*Goldberg et al., 2019*). This suggests an important role for mitochondrial fatty acid oxidation and hepatic ketogenesis in protection against viral infections.

Our findings assume strong clinical significance as recent studies demonstrate that γδ T cells were severely depleted in COVID-19 patients in two highly variable cohorts, and disease progression was correlated with near ablation of Vγ9Vδ2 cells that are dominant subtype of circulating γδ T cells (*Laing et al., 2020*). Taken together these data demonstrate that a ketogenic immunometabolic switch protects against mCoV-A59-driven infection in mice and this involves anti-inflammatory response in lung which is coupled with reduction of NLRP3 inflammasome, restoration of protective ϒδ T cells, and remodeling of the pool of the inflammatory monocytes. Finally, our results suggest that acutely switching infected or at-risk elderly patients to a KD may ameliorate COVID-19 and, therefore, is a relatively accessible and affordable intervention that can be promptly applied in most clinical settings.

## Materials and methods

### Key resources table

| Reagent type (species) or resource | Designation | Source or reference | Identifiers | Additional information |
|---|---|---|---|---|
| Strain, strain background (mouse, male) | C57BL/6 | NIA | | |
| Strain, strain background (Murine Coronavirus) | MHV-A59 | Bei resources | NR-43000 | |
| Cell line (*Rattus norvegicus*) | L2 | ATCC | CCL-149 | |
| Cell line (*Mus musculus*) | L929 | ATCC | CCL1 | |
| Biological sample (*Mus musculus*) | Primary BMDM | This paper | | Freshly isolated from mouse |
| Antibody | Rat anti-Caspase-1 | Genentech | N/A | WB (1:250) |
| Antibody | Rabbit polyclonal anti-β actin | Cell Signaling Technology | Cat#4967L; RRID:AB_330288 | WB (1:1000) |
| Antibody | Rabbit polyclonal anti-IL-1β | GeneTex | Cat#GTX74034; RRID:AB_378141 | WB (1:1000) |

*Continued on next page*

*Continued*

| Reagent type (species) or resource | Designation | Source or reference | Identifiers | Additional information |
|---|---|---|---|---|
| Antibody | Rabbit polyclonal anti-ASC | AdipoGen | Cat#AG-25B-0006; RRID:AB_2490440 | WB (1:1000) |
| Antibody | BV711 Rat monoclonal anti-CD45 | BioLegend | Cat#103147; RRID:AB_2564383 | FACS (1:100) |
| Antibody | FITC Rat monoclonal anti-MERTK | BioLegend | Cat#151504; RRID:AB_2617035 | FACS (1:100) |
| Antibody | BV605 Mouse monoclonal anti-CD64 | BioLegend | Cat#139323; RRID:AB_2629778 | FACS (1:100) |
| Antibody | eFluor 450 Rat monoclonal anti-F4/80 | Thermo Fisher Scientific | Cat#48-4801-82; RRID:AB_1548747 | FACS (1:100) |
| Antibody | PerCP-Cy5.5 Rat monoclonal anti-Ly-6C | Thermo Fisher Scientific | Cat#45-5932-82; RRID:AB_2723343 | FACS (1:100) |
| Antibody | APC Hamster monoclonal anti-CD11c | BioLegend | Cat#117310; RRID:AB_313779 | FACS (1:100) |
| Antibody | PE Rat monoclonal anti-CD169 | Thermo Fisher Scientific | Cat#12-5755-82; RRID:AB_2572625 | FACS (1:100) |
| Antibody | PE/Cy7 Rat monoclonal anti-CD86 | BioLegend | Cat#105014; RRID:AB_439783 | FACS (1:100) |
| Antibody | BV605 Rat monoclonal anti-CD3 | BioLegend | Cat#100237; RRID:AB_2562039 | FACS (1:100) |
| Antibody | PE-Cy7 Rat monoclonal anti-CD4 | Thermo Fisher Scientific | Cat#25-0042-82; RRID:AB_469578 | FACS (1:100) |
| Antibody | eFluor 450 Rat monoclonal anti-CD8a | Thermo Fisher Scientific | Cat#48-0081-82; RRID:AB_1272198 | FACS (1:100) |
| Antibody | PE Hamster monoclonal anti-TCR γ/δ | BioLegend | Cat#118108; RRID:AB_313832 | FACS (1:100) |
| Antibody | APC Rat monoclonal anti-Ly-6G | Thermo Fisher Scientific | Cat#17-9668-82; RRID:AB_2573307 | FACS (1:100) |
| Antibody | PerCP-Cy5.5 Rat monoclonal anti-Siglec-F | BD Biosciences | Cat#565526; RRID:AB_2739281 | FACS (1:100) |
| Antibody | PerCP-Cy5.5 Rat monoclonal anti-CD62L | BioLegend | Cat#104432; RRID:AB_2285839 | FACS (1:100) |
| Antibody | APC/Cy7 Rat monoclonal anti-CD44 | BioLegend | Cat#103028; RRID:AB_830785 | FACS (1:100) |
| Antibody | Rat monoclonal anti-CD68 | Bio-Rad | Cat#MCA1957; RRID:AB_322219 | IF (1:100) |
| Peptide, recombinant protein | Recombinant Mouse M-CSF Protein | R and D Systems | Cat#416 ML-050 | |

*Continued on next page*

*Continued*

| Reagent type (species) or resource | Designation | Source or reference | Identifiers | Additional information |
|---|---|---|---|---|
| Commercial assay or kit | RNeasy Plus micro kit | Qiagen | Cat#74034 | |
| Commercial assay or kit | Direct-zol RNA Miniprep Plus kit | Zymo Research | Cat#R2072 | |
| Commercial assay or kit | iScript cDNA synthesis kit | Bio-Rad | Cat#1708891 | |
| Commercial assay or kit | Power SYBR Green PCR Master Mix | Thermo Fisher Scientific | Cat#4367659 | |
| Commercial assay or kit | ProcartaPlex multiplex assay | Thermo Fisher Scientific | Cat#PPX | |
| Chemical compound, drug | Alexa Fluor 594 Phalloidin | Thermo Fisher Scientific | Cat#A12381, RRID:AB_2315633 | |
| Chemical compound, drug | LPS | Sigma | Cat#L3024 | |
| Chemical compound, drug | Pam3CSK4 | Invivogen | Cat#tlrl-pms | |
| Chemical compound, drug | (R)-Hydroxybutyric Acid (BHB) | Sigma | Cat#54920–1 G-F | |
| Chemical compound, drug | Disuccinimidyl suberate (DSS) | Thermo Fisher Scientific | Cat#21655 | |
| Software, algorithm | Image J | NIH | https://imagej.nih.gov/ij/ | |
| Software, algorithm | Prism 7 | Graphpad | https://www.graphpad.com/ | |
| Software, algorithm | FlowJo | Treestar | https://www.flowjo.com/ | |
| Software, algorithm | Cell Ranger Single-Cell Software Suite (v3.0.2) | 10x genomics | https://support.10xgenomics.com | |
| Software, algorithm | R (v3.5.0) package Seurat (v3.1.1) | *Butler et al., 2018* | https://cran.r-project.org/web/packages/Seurat/index.html | |
| Software, algorithm | MAST | *Finak et al., 2015* | https://github.com/RGLab/MAST | |
| Software, algorithm | clusterProfiler package (v3.12.0) | *Yu et al., 2012* | http://bioconductor.org/packages/release/bioc/html/clusterProfiler.html | |
| Software, algorithm | STAR (v2.7.3a) | *Dobin et al., 2013* | http://code.google.com/p/rna-star/ | |
| Software, algorithm | MultiQC (v1.9) | *Ewels et al., 2016* | http://multiqc.info | |
| Software, algorithm | HTSeq framework (v0.11.2) | *Anders et al., 2015* | http://www-huber.embl.de/HTSeq | |

*Continued on next page*

*Continued*

| Reagent type (species) or resource | Designation | Source or reference | Identifiers | Additional information |
|---|---|---|---|---|
| Software, algorithm | DeSeq2 package (v1.24.0) | *Love et al., 2014* | http://www. bioconductor. org/packages/release/ bioc/html/ DESeq2.html | |
| Software, algorithm | fgsea R package (v1.10.0) | *Sergushichev, 2016* | https://github. com/ctlab/fgsea | |

## Study design

The object of this study was to investigate whether KD affects defense response of mice against MHV-A59 along with identification of underlying mechanism. Mice were randomized for all experiments groups. For experiments with mice provided with different diets, mice were fed with chow diet or KD for 5 days before infection following phenotype observation. It was not possible that investigators were blinded for KD experiments because appearance of KD was different from chow diet, while investigators were blinded for experiments and data analysis. Number of samples are indicated in each figure legend. Sample size for experiments was decided based on power calculation from previous experiments and experiences. No data points were excluded in this study. All experiments were repeated independently at least two times. Bulk RNA-sequencing experiment was performed once with biologically independent samples. Single-cell RNA-sequencing experiment was done once with pooling biologically independent samples per group.

## Animal models

All mice used in this study were C57BL/6 mice. Old mice (20–24 month old) were received from NIA, maintained in our laboratory. Young mice (2–6 months old) were from NIA or purchased from Jackson Laboratories or bred in our laboratory. The mice were housed in specific pathogen-free facilities with free access to sterile water through Yale Animal Resources Center. Mice were fed a standard vivarium chow (Harlan 2018s) or a ketogenic diet (Envigo, TD.190049) for indicated time. The mice were housed under 12 hr light/dark cycles. All experiments and animal use were conducted in compliance with the National Institute of Health Guide for the Care and Use of Laboratory Animals and were approved by the Institutional Animal Care and Use Committee (IACUC) at Yale University.

## Viral infection

MHV-A59 was purchased from Bei resources (NR-43000) and grown in BV2 cells. Mice were anesthetized by intraperitoneal injection of ketamine/xylazine. 700, 7000, or $10^6$ PFU of MHV-A59 was delivered in 40 μl PBS via intranasal inoculation. Vital signs were measured before and after infection. Arterial oxygen saturation, breath rate, heart rate, and pulse distention were measured in conscious, unrestrained mice via pulse oximetry using the MouseOx Plus (Starr Life Sciences Corp.).

## Electron microscopy

Lungs were fixed in 10% formaldehyde, osmicated in 1% osmium tetroxide, and dehydrated in ethanol. During dehydration, 1% uranyl acetate was added to the 70% ethanol to enhance ultrastructural membrane contrast. After dehydration, the lungs were embedded in Durcupan, and ultrathin sections were cut on a Leica Ultra-Microtome, collected on Formvar-coated single-slot grids, and analyzed with a Tecnai 12 Biotwin electron microscope (FEI).

## Histology and immunohistochemistry

H and E and MSB staining of lung tissues were performed on sections of formalin-fixed paraffin-embedded at the Comparative Pathology Research core at Yale School of Medicine. For immunohistochemistry, the hearts were harvested from MHV-A59-infected mice, fixed in 4% PFA overnight, and embedded in OCT after dehydration with 30% sucrose, and serial sections of aortic root were cut at 6 μm thickness using a cryostat. Sections were incubated at 4°C overnight with CD68 (Serotec;

#MCA1957) and Alexa Fluor 594 Phalloidin (ThermoFisher, A12381) after blocking with blocker buffer (5% Donkey Serum, 0.5% BSA, 0.3% Triton X-100 in PBS) for 1 hr at RT, followed by incubation with Alexa Fluor secondary antibody (Invitrogen, Carlsbad, CA) for 1 hr at RT. The stained sections were captured using a Carl Zeiss scanning microscope Axiovert 200M imaging system, and images were digitized under constant exposure time, gain, and offset. Results are expressed as the percent of the total plaque area stained measured with the Image J software (ImageJ version 1.51).

## Plaque assay

L2 cell (1.5 ml of $6 \times 10^5$ cells/ml) were seeded on six-well plates (Corning, 3516) in supplemented DMEM and allowed to adhere overnight. Tissue samples were homogenized in unsupplemented DMEM and spun down at 2000 rpm for 5 min. Supernatant was serially diluted and 200 µL of each sample was added to aspirated L2 cells in six-well plates. Plates were agitated regularly for 1 hr before adding overlay media consisting of 1 part 1.2% Avicel and 1 part 2× DMEM (Thermo Fisher, 12800) supplemented with 4% FBS (Thermo Fisher, A3840301), penicillin-streptomycin (Gibco, 15140122), MEM Non-Essential Amino Acids Solution (Gibco, 11140050), and HEPES (Gibco, 15630080). After a 4-day incubation, cells were fixed in 10% formaldehyde (Sigma-Aldrich, 8187081000) diluted with PBS for 1 hr. Cells were then stained in 1% (w/v) crystal violet (Sigma Aldrich, C0775) for 1 hr, washed once in distilled water, and then quantified for plaque formations.

## Multiplex cytokine analyses

Serum cytokine and chemokine level was measured by ProcartaPlex multiplex assay (Thermo Fisher Scientific). Assay was prepared following manufacture's instruction. Twenty-five microliters of collected serum from each mice in this study was used. Customized assay including IL-1β, TNFα, IL-6, MCP-1, and MIP-1β was used. Luminex xPONENT system was used to perform the assay.

## Quantitative PCR

To extract and purify RNA from tissues, RNeasy Plus micro kit (Qiagen) and Direct-zol RNA Miniprep Plus kit (Zymo Research) were used according to manufacturer's instructions. cDNA was synthesized with isolated RNA using iScript cDNA synthesis kit (Bio-Rad). To quantify amount of mRNA, real-time quantitative PCR was done with the synthesized cDNA, gene-specific primers, and Power SYBR Green detection reagent (Thermo Fischer Scientific) using the LightCycler 480 II (Roche). Analysis was done by ΔΔCt method with measured values from specific genes, the values were normalized with *Gaphd* gene as an endogenous control. Primer information is described in *Supplementary file 1*.

## BMDM culture and in vitro viral infection

BMDM was cultured by collecting mouse femurs and tibias in complete collecting media containing RPMI (Thermo Fischer Scientific), 10% FBS (Omega Scientific), and 1% antibiotics/antimycotic (Thermo Fischer Scientific). Using needle and syringe, bone marrow was flushed into new complete media, followed by red blood cells lysis by ACK lyses buffer (Quality Biological). In six-well plate, the collected cells were seeded to be differentiated into macrophages incubated with 10 ng/ml M-CSF (R and D) and L929 (ATCC) conditioned media. Cells were harvested on day 7 and seeded as $1 \times 10^6$ cell/well in 24-well plate for experiments. To infect BMDM, MHV-A59 was incubated with BMDM as a MOI 1 (1:1) for 24 hr. For inflammasome activation, LPS (1 µg/ml) or Pam3CSK4 (1 µg/ml) was pre-treated with or without BHB (1, 5, and 10 mM) for 4 hr before MHV-A59 infection for 24 hr or ATP (5 mM) treatment for 1 hr.

## Western blotting

To prepare samples for western blotting, tissues were snap frozen in liquid nitrogen. RIPA buffer with protease inhibitors was used to homogenize the tissues. After cell supernatant was collected, cells were harvested by directly adding RIPA buffer on cell culture plate. After quantification of protein amount by the DC protein assay (Bio-Rad), the same amount of protein was run on SDS–PAGE gel followed by transferring to nitrocellulose membrane. Specific primary antibodies and appropriate secondary antibodies (Thermo Fisher Scientific) were used to probe blots, and bands were detected by ECL Western Blotting Substrate (Pierce). The following primary antibodies were used for

experiments: Antibodies to caspase-1 (1:250, Genentech), β-actin (1:1,000, 4967L; Cell Signaling), IL-1β (1:1000, GTX74034, GeneTex), and ASC (1:1000, AG-25B-0006, AdipoGen).

## ASC oligomerization assay

To detect ASC oligomers, cells were harvested in NP-40 lysis buffer that contains 20 mM HEPES–KOH (pH 7.5), 150 mM KCl, 1% NP-40, 0.1 mM PMSF, and protease inhibitors. The cells in lysis buffer were incubated on ice for 15 min and centrifuged at 6000 rpm at 4°C for 10 min. Supernatant was collected and kept for cell lysate western blotting. The pellet was vortexed with 1 ml of NP-40 lysis buffer and centrifuged at 6000 rpm at 4°C for 10 min. The pellet was incubated with 50 µl of NP-40 lysis buffer and 1 µl of 200 mM disuccinimidyl suberate for 30 min at room temperature and then centrifuged at 6000 rpm at 4°C for 10 min. The pellet with SDS sample buffer and reducing reagent was loaded for western blotting.

## Flow cytometry

Lung was digested in RPMI (Thermo Fisher) with 0.5 mg/ml Collagenase I (Worthington) and 0.2 mg/ml DNase I (Roche) for 1 hr. Digested lung tissues were minced through 100 µm strainer. Spleen was directly minced through 100 µm strainer. Minced tissues were additionally filtered with 40 µm strainer after red blood cell lysis by ACK lysing buffer (Quality Biological). After incubation with Fc Block CD16/32 antibodies (Thermo Fisher Scientific), the cells from lung and spleen were further incubated with surface antibodies for 30 min on ice in the dark. Washed cells were stained with LIVE/DEAD Fixable Aqua Dead Cell Stain Kit (Thermo Fisher Scientific). BD LSRII was used for flow cytometry, and results were analyzed by FlowJo software. The following antibodies were used for flow cytometry analysis to detect CD4 T cell, CD8 T cell, γδ T cell, neutrophil, eosinophil, and macrophage: CD45-BV711, MerTK-FITC, CD64-BV605, F4/80-eFluor450, Ly6C-PerCP-Cy5.5, CD11c-APC, CD169-PE, CD86-PE-Cy7, CD3-BV605, CD4-PE-Cy7, CD8-eFluor450, TCR γ/δ-PE, Ly6G-APC, SiglecF-BV605, CD62L-PerCP-Cy5.5, CD44-APC-Cy7.

## Single-cell RNA-sequencing

Lung cells were prepared as mentioned above for flow cytometry and equal amount of cells were pooled as indicted in the experiments. Single-cell RNA-sequencing libraries were prepared at Yale Center for Genome Analysis following manufacturer's instruction (10× Genomics). NovaSeq6000 was used for sequencing library read.

## Alignment, barcode assignment, and unique molecular identifier counting

The Cell Ranger Single-Cell Software Suite (v3.0.2) (available at https://support.10xgenomics.com/single-cell-gene-expression/software/pipelines/latest/what-is-cell-ranger) was used to perform sample demultiplexing, barcode processing, and single-cell 3' counting. Cellranger mkfastq was used to demultiplex raw base call files from the NovaSeq6000 sequencer into sample-specific fastq files. Subsequently, fastq files for each sample were processed with cellranger counts to align reads to the mouse reference (version mm10-3.0.0) with default parameters.

## Preprocessing analysis with Seurat package

For the analysis, the R (v3.5.0) package Seurat (v3.1.1) (*Butler et al., 2018*) was used. Cell Ranger filtered genes by barcode expression matrices were used as analysis inputs. Samples were pooled together using the merge function. The fraction of mitochondrial genes was calculated for every cell, and cells with high (>5%) mitochondrial fraction were filtered out. Expression measurements for each cell were normalized by total expression and then scaled to 10,000, after that log normalization was performed (NormalizeData function). Two sources of unwanted variation – unique molecular identifier (UMI) counts and fraction of mitochondrial reads – were removed with ScaleData function. For both datasets, platelet clusters as well as a cluster of degraded cells (no specific signature and low UMI count) were removed and data was re-normalized without them. In case of Old-Keto and Old-Chow dataset, we additionally removed neutrophils, doublets, and red blood cells. Doublets were manually defined as cells that were positive for two or more marker sets.

### Dimensionality reduction and clustering

The most variable genes were detected using the FindVariableGenes function. Principal components analysis (PCA) was run only using these genes. Cells are represented with >Uniform Manifold Approximation and Projection (UMAP) plots. We applied RunUMAP function to normalized data, using first 15 PCA components. For clustering, we used functions FindNeighbors and FindClusters that implement shared nearest neighbor modularity optimization-based clustering algorithm on top 20 PCA components using resolution of 0.3 for both datasets.

### Identification of cluster-specific genes and marker-based classification

To identify marker genes, FindAllMarkers function was used with likelihood-ratio test for single-cell gene expression. For each cluster, only genes that were expressed in more than 10% of cells with at least 0.1-fold difference (log-scale) were considered. For heatmap representation, mean expression of markers inside each cluster calculated by AverageExpression function was used.

### Single-cell RNA-seq differential expression

To obtain differential expression between clusters, MAST test was performed via FindMarkers function on genes expressed in at least 1% of cells in both sets of cells, and p-value adjustment was done using a Bonferroni correction (*Finak et al., 2015*). Pathway analysis was performed using clusterProfiler package (v3.12.0) (*Yu et al., 2012*), with Hallmark gene sets from MSigDB. Significantly different genes were used ($p_{adj}<0.05$) if percent difference between conditions (|pct.1 – pct.2|) was over 1%. To visualize pathway expression for each cell z-scores of all pathways, genes were averaged.

### Cell subsets analysis

To separate αβ T cells and γσ T cells, we subset raw values of T cell (cluster 2), normalized and clustered corresponding data separately as described above with clustering resolution 0.2. Obtained subclusters were projected on the original UMAP, splitting cluster 2 in 2_1 (αβ T cells) and 2_2 (γσ T cells). Monocyte clusters 0 and 1 were subset and re-analyzed in the same manner with clustering resolution 0.3. UMAP was recalculated for monocytes only as shown in *Figure 6F*.

### Bulk RNA-sequencing of sorted γδ T cells

Cells from lung were prepared as mentioned above for flow cytometry, and γδ T cell was sorted by flow cytometry (live CD45+CD3+CD4 CD8− TCR γ/δ+). RNA was isolated from sorted cells using RNeasy Plus micro kit (Qiagen). Quality checked RNA was used for RNA-sequencing library preparation at Yale Center for Genome Analysis following manufacturer's instruction (Illumina). NovaSeq6000 was used for sequencing library read.

Fastq files for each sample were aligned to the mm10 genome (Gencode, release M25) using STAR (v2.7.3a) with the following parameters: STAR –`genomeDir` $GENOME_DIR –`readFilesIn` $WORK_DIR/$FILE_1 $WORK_DIR/$FILE_2 –`runThreadN` 12 –`readFilesCommand` zcat –`outFilterMultimapNmax` 15 –`outFilterMismatchNmax` 6 –`outReadsUnmapped` Fastx –`outSAMstrandField` intronMotif –`outSAMtype` BAM SortedByCoordinate –`outFileNamePrefix`. /$ (*Dobin et al., 2013*). Quality control was performed by FastQC (v0.11.8), MultiQC (v1.9) (*Ewels et al., 2016*), and Picard tools (v2.21.6). Quantification was done using htseq-count function from HTSeq framework (v0.11.2): htseq-count -f bam -r pos -s no -t exon $BAM $ANNOTATION > $OUTPUT (*Anders et al., 2015*). Differential expression analysis was done using DESeq function from DeSeq2 package (*Love et al., 2014*) (v1.24.0) with default settings. Significance threshold was set to adjusted p-value<0.05. Gene set enrichment analysis via fgsea R package (*Sergushichev, 2016*) (v1.10.0) was used to identify enriched pathways and plot enrichment curves.

### Statistical analysis

To calculate statistical significance, two-tailed Student's t-test was used. Level of significance was indicated as follows: *p<0.05; **p<0.005; ***p<0.001; ****p<0.0001. All statistical tests used 95% confidence interval, and normal distribution of data was assumed. Biological replication numbers for each experiment were indicated in each figure and figure legend. Data were shown as mean ± S.E.M. GraphPad Prism software was used for all statistical tests to analyze experimental results.

## Acknowledgements

SR is supported by American Federation of Aging Research (AFAR) postdoctoral fellowship award. The Dixit lab is supported in part by NIH grants P01AG051459, AR070811 and Cure Alzheimer's Fund. The Wang lab is supported in part by NIH grant 1K08AI128745 and by gifts from the Knights of Columbus, G Harold and Leila Y Mathers Charitable Foundation, and the Ludwig Family Foundation. We also thank the Yale Center on Genomic Analysis (YCGA) for RNA-seq studies and Genentech Inc for providing the anti-caspase-1 antibody.

## Additional information

### Funding

| Funder | Grant reference number | Author |
|---|---|---|
| National Institute on Aging | P01AG051459 | Vishwa Deep Dixit |
| National Institute of Arthritis and Musculoskeletal and Skin Diseases | AR070811 | Vishwa Deep Dixit |
| American Federation for Aging Research | Glenn Foundation for Medical Research Postdoctoral Fellowships in Aging Research | Seungjin Ryu |
| National Institute of Allergy and Infectious Diseases | 1K08AI128745 | Andrew Wang |
| G. Harold and Leila Y. Mathers Charitable Foundation | MF200400809 | Andrew Wang |
| Ludwig Family Foundation | | Andrew Wang |
| Knights of Columbus | | Andrew Wang |

The funders had no role in study design, data collection and interpretation, or the decision to submit the work for publication.

### Author contributions

Seungjin Ryu, Formal analysis, Investigation, Writing - original draft, Writing - review and editing; Irina Shchukina, Data curation, Formal analysis, Writing - review and editing; Yun-Hee Youm, Investigation, Performed experiments regarding mouse tissue processing and qPCR; Hua Qing, Investigation, Writing - review and editing, Performed infections and phenotyping of mice; Brandon Hilliard, Investigation, Performed plaque assay; Tamara Dlugos, Investigation, Performed experiments regarding mouse tissue processing; Xinbo Zhang, Investigation, Performed experiments about impact of infection on heart; Yuki Yasumoto, Investigation, Executed electron microscopy; Carmen J Booth, Investigation, Performed the histological analyses; Carlos Fernández-Hernando, Yajaira Suárez, Resources, Supervision, Writing - review and editing; Kamal Khanna, Conceptualization, Writing - review and editing, Designed lung immune cell analyses experiments; Tamas L Horvath, Resources, Supervision, Writing - review and editing, Designed electron microscopy; Marcelo O Dietrich, Resources, Supervision, Writing - review and editing, Designed experiments and analyzed hypothalamic inflammation; Maxim Artyomov, Resources, Formal analysis, Supervision, Writing - review and editing; Andrew Wang, Conceptualization, Resources, Supervision, Writing - review and editing; Vishwa Deep Dixit, Conceptualization, Supervision, Funding acquisition, Writing - original draft, Writing - review and editing

### Author ORCIDs

Seungjin Ryu https://orcid.org/0000-0001-6353-8789
Tamas L Horvath http://orcid.org/0000-0002-7522-4602
Marcelo O Dietrich http://orcid.org/0000-0001-9781-2221
Andrew Wang https://orcid.org/0000-0002-6951-8081
Vishwa Deep Dixit https://orcid.org/0000-0002-5341-6494

## Ethics

Animal experimentation: All experiments and animal use were conducted in compliance with the National Institute of Health Guide for the Care and Use of Laboratory Animals and were approved by the Institutional Animal Care and Use Committee (IACUC) protocol (#2019-11572 and 2020-20149) of Yale University.

## Decision letter and Author response

Decision letter https://doi.org/10.7554/eLife.66522.sa1
Author response https://doi.org/10.7554/eLife.66522.sa2

# Additional files

## Supplementary files

• Supplementary file 1. Primer information for qPCR. The table includes information about forward and reverse primers for genes used in qPCR experiments.

• Transparent reporting form

## Data availability

The single cell RNA-sequencing and bulk RNA-sequencing data has been uploaded to Gene Expression Omnibus, GSE155346 and GSE155347, respectively.

The following datasets were generated:

| Author(s) | Year | Dataset title | Dataset URL | Database and Identifier |
|---|---|---|---|---|
| Ryu S, Shchukina I, Youm YH, Qing H, Hilliard B, Dlugos T, Zhang X, Yasumoto Y, Booth CJ, Fernández-Hernando C, Suárez Y, Khanna K, Horvath TL, Dietrich MO, Artyomov M, Wang A, Dixit VD | 2020 | Ketogenesis restrains aging-induced exacerbation of COVID in a mouse model | https://www.ncbi.nlm.nih.gov/geo/query/acc.cgi?acc=GSE155346 | NCBI Gene Expression Omnibus, GSE155346 |
| Ryu S, Shchukina I, Youm YH, Qing H, Hilliard B, Dlugos T, Zhang X, Yasumoto Y, Booth CJ, Fernández-Hernando C, Khanna K, Horvath TL, Dixit VD, Dietrich MO, Artyomov M, Wang A | 2020 | Ketogenesis restrains aging-induced exacerbation of COVID in a mouse model | https://www.ncbi.nlm.nih.gov/geo/query/acc.cgi?acc=GSE155347 | NCBI Gene Expression Omnibus, GSE155347 |

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
