## [Decision Letter]

**Acceptance summary:**

This study used an aging model of natural murine beta coronavirus infection to demonstrate a protective effect of ketogenic diet on viral infection through expanding protective γδ T cells and decreasing myeloid cell-mediated inflammatory responses. As this model recapitulates several hallmarks of human COVID-19, the findings of this study support the use of ketogenic diet as a potential treatment against SARS-CoV-2 infection for older adults who are more vulnerable to this disease.

**Decision letter after peer review:**

Thank you for submitting your article "Ketogenic diet restrains aging-induced exacerbation of coronavirus infection in mice" for consideration by *eLife*. Your article has been reviewed by 3 peer reviewers, and the evaluation has been overseen by a Reviewing Editor and Tadatsugu Taniguchi as the Senior Editor. The following individual involved in review of your submission has agreed to reveal their identity: Berislav Bosnjak (Reviewer #2).

Essential revisions:

1. The flow cytometry data presented in figure 1 is better understood with the entirety of the data. For example, CD4 T cells in uninfected and infected old mice are at similar levels, however as presented the data in Figure 1 only show infected young v infected old with a significant reduction. Include the Figure 1 supplementary data B-D in Figure 1 and include these complexities in the Results section. Viral load should also be included in the main figure.

2. The authors should include the absolute cell numbers (Figure 1F-J) and indicate whether these data were from lung or spleen. In addition, please comment on whether Treg were examined in the CD4 T cell population.

3. Similar comment for Figure 3 as old mice have elevated IL1-β already, so the difference is not necessarily due to infection, but old age. Include supplementary data in Figure 3 instead of the partial data shown and explain in Results section.

4. The authors should provide data on whether the ketogenic diet saved the mice from mortality due to infection (Figure 4).

5. The authors should comment on whether the reductions in inflammatory mediators identified in Figure 5 were equivalent to those observed in young mice.

6. The authors should comment on which Vγ chains were more prevalent in the infected aged vs young populations, as different γδ T cell populations play differing roles in the lung (Figure 1G, 5G). In addition, please comment on any cytokine genes were noted in γδT cell RNA-seq data (Figure 6D)

7. The gating strategy for lung leukocyte populations depicted in Figure 1—figure supplement is not complete and unclear. It seems that cells positive for live/dead staining were included in the analysis (as depicted in Figure 1—figure supplement E and H). Moreover, the authors do not show gating for all populations included in the study (e.g. eosinophils or neutrophils). The same applies to the data shown in Figure 5—figure supplement. Please re-adjust the gating strategy, assuring that all dead cells are excluded from the analysis. Further, it is crucial that the gating strategy is clearly depicted for all investigated subsets.

8. Figure 5 and Figure 1—figure supplement: as for figure 1, please express the data as cell numbers rather than as cell percentages.

9. it is not clear whether the authors compare aged to adult mice without or with infection. Please re-phrasing the paragraph (Line 106-113) and clearly stating differences (i) induced by aging and (ii) by an infection in each age group.

10. Please comment on whether the effect observed with a ketogenic diet could be mimicked by BHB alone.

11. Regarding single-cell RNA sequencing analysis: (i) are there any differences in monocytes between adult and aged mice without infection? (ii) what are the differences between uninfected and infected mice of the same age group? (iii) what are the differences in gene expression per cell type when aged mice fed with normal or KD are compared? In other words, are there any common changes induced by KD feeding that are common to all cell types? (iv) please specify the number of cells per sample that were analyzed and the sequencing depth. Why were neutrophils excluded from the "Old-Keto and Old-Chow dataset"? How were the doublets detected? (v) Why was a % of mitochondrial genes set at 5%, as it is possible that highly activated cells have higher expression of mitochondrial genes? Does the KD have an effect on mitochondrial gene expression?

12. Did the authors also analyze with single cell-RNA sequencing of infected young mice? Do those mice also miss cluster 1 monocyte population (referring to Figure 6F)? This would imply the relative importance of those cells in the immunopathology of the lungs in aged mice. Further, would it be possible to characterize the monocytes from cluster 1 with extracellular markers within the flow cytometry data?

---

## [Author Response]

Essential revisions:1. The flow cytometry data presented in figure 1 is better understood with the entirety of the data. For example, CD4 T cells in uninfected and infected old mice are at similar levels, however as presented the data in Figure 1 only show infected young v infected old with a significant reduction. Include the Figure 1 supplementary data B-D in Figure 1 and include these complexities in the Results section. Viral load should also be included in the main figure.

We sincerely thank the Reviewer for the suggestion. In the revised Figure 1, we included the data in Figure 1—figure supplement 1B-D to show all data to include uninfected and infected young and old mice (now Figure 1G-L). The viral load data from Figure 1—figure supplement 1A was also included to improve clarity (now Figure 1F). We also included the description of the updated figures in Results section of manuscript (Line 106-124).

2. The authors should include the absolute cell numbers (Figure 1F-J) and indicate whether these data were from lung or spleen. In addition, please comment on whether Treg were examined in the CD4 T cell population.

We thank the Reviewer for the suggestion. We additionally included the flow cytometry data indicated as absolute cell numbers for Figure 1F-J (now Figure 1G-L) and clearly indicated that the cells were derived from lungs in figure 1 and in the legend. We included the interpretation of the data in Results section of manuscript (Line 106-124).

We also thank the Reviewer for the suggestion about clarifying whether CD4 T cell population analyses included analyses of Tregs. Tregs are known to increase with aging in spleen and adipose tissue (Bapat et al., 2015, PMID: 26580014). As suggested by the Reviewers, the Treg population could be identified by additional scRNA-seq analysis with young and old infected mice. These analyses revealed that Treg cells in lung between young and old infected mice were not significantly different and comparisons of fraction of Foxp3 positive T cell population are included now in Figure 6—figure supplement 1E-G. The manuscript is revised to include these data in Results section (Line 272-273).

3. Similar comment for Figure 3 as old mice have elevated IL1-β already, so the difference is not necessarily due to infection, but old age. Include supplementary data in Figure 3 instead of the partial data shown and explain in Results section.

We thank the Reviewer for the suggestion. Indeed, we and others have shown that Nlrp3 inflammasome-dependent IL-1β increases with aging. As per Reviewer’s suggestion, we have included the Western blot results (Figure 3I). We agree that basal inflammasome activation increases in aging and show that it is further exacerbated post infection; higher transcriptional levels of Casp1 and Nlrp3 in VAT from old infected mice compared to old uninfected mice support the results (Figure 3K and Figure 3—figure supplement 1K). We have revised the manuscript by including these new details in Results section (Line 165-168).

4. The authors should provide data on whether the ketogenic diet saved the mice from mortality due to infection (Figure 4).

As suggested by the Reviewer, we performed additional experiments to examine the impact of ketogenesis on coronavirus infection-induced mortality in both young and old mice upon infection with lethal doses of MHV (PFU 1e6). These results show that ketogenic diet-fed young mice are partially protected from death post infection; however, KD is unable to rescue old mice from high dose MHV-A59 viral infection-induced death. We have included these data in Figure 4 (H-J). These data are an important addition to the manuscript that shows that the role of ketogenesis in protecting old mice against disease severity and recovery but not in mortality caused by lethal infection. We included the description of new data in Results section (Line 228-235).

5. The authors should comment on whether the reductions in inflammatory mediators identified in Figure 5 were equivalent to those observed in young mice.

Our work primarily focused on aging. We have included new data that KD-fed young mice are protected from infection-induced lethality while KD-fed old mice are not (Figure 4I, J). Also, we have included data on the impact of MHV-A59 on inflammatory gene expression in young and old mice in visceral adipose tissue (VAT) and hypothalamus (Figure 3K and Figure 3—figure supplement 1H-Q). These results show that infection significantly increases *Casp1*, *Tnf*, *Nlrp3* and *Il6* expression in VAT and *Casp1* and *Tnf* in hypothalamus.

6. The authors should comment on which Vγ chains were more prevalent in the infected aged vs young populations, as different γδ T cell populations play differing roles in the lung (Figure 1G, 5G). In addition, please comment on any cytokine genes were noted in γδT cell RNA-seq data (Figure 6D)

As suggested by the Reviewer, we re-analyzed the subtypes of γ chain and cytokines represented in the RNA-sequencing analysis of γδT cell in aged mice on chow and ketogenic diet. Compared to chow, there is no significant difference in representation of Vγ chains in the γδT cells expanded by ketogenic data. We have included these data in Figure 6—figure supplement 2D, E. and it was described in Results section (Line 284-286).

Also, *Ifng* and *Il17a* expression was examined by RNA-sequencing of γδT cells in aged mice on chow and ketogenic diet. These data also show that ketogenic diet does not significantly change the transcription of cytokine genes. These data are now included in Figure 6—figure supplement 2F and it was described in Results section (Line 284-286).

7. The gating strategy for lung leukocyte populations depicted in Figure 1—figure supplement is not complete and unclear. It seems that cells positive for live/dead staining were included in the analysis (as depicted in Figure 1—figure supplement E and H). Moreover, the authors do not show gating for all populations included in the study (e.g. eosinophils or neutrophils). The same applies to the data shown in Figure 5—figure supplement. Please re-adjust the gating strategy, assuring that all dead cells are excluded from the analysis. Further, it is crucial that the gating strategy is clearly depicted for all investigated subsets.

We appreciate the Reviewer’s remarks and have accordingly re-analyzed and corrected/clarified the gating and, in particular, the live/dead gating plots. We have accordingly revised the manuscript and revised the gating strategy figures in Figure 1—figure supplement 1 and Figure 5—figure supplement 1, 2. In addition, gating plots for eosinophils and neutrophils were also included in the gating schematic in Figure 1—figure supplement 1 and Figure 5—figure supplement 1 for clarity.

8. Figure 5 and Figure 1—figure supplement: as for figure 1, please express the data as cell numbers rather than as cell percentages.

As suggested by the Reviewer, he have revised the manuscript and included the absolute cell numbers in Figure 5I, Figure 5—figure supplement 1, 2, and Figure 1—figure supplement 1.

9. It is not clear whether the authors compare aged to adult mice without or with infection. Please re-phrasing the paragraph (Line 106-113) and clearly stating differences (i) induced by aging and (ii) by an infection in each age group.

We thank the Reviewer for the suggestion. Together with figure changes made in response to #1, we have edited the paragraph to clearly indicate that the old infected mice were compared to young infected as well as old uninfected mice. We included the description of the updated figures in Results section of manuscript (Line 106-124).

10. Please comment on whether the effect observed with a ketogenic diet could be mimicked by BHB alone.

In a previous study in our lab (Goldberg et al., 2019, PMID: 31732517), we showed that increasing BHB by feeding ketone esters—bypassing hepatic ketogenesis and mitochondrial fatty acid oxidation—is not sufficient to confer protection against influenza infection. We have included this important point in the Discussion section (Line 371-375).

11. Regarding single-cell RNA sequencing analysis: (i) are there any differences in monocytes between adult and aged mice without infection? (ii) what are the differences between uninfected and infected mice of the same age group? (iii) what are the differences in gene expression per cell type when aged mice fed with normal or KD are compared? In other words, are there any common changes induced by KD feeding that are common to all cell types? (iv) please specify the number of cells per sample that were analyzed and the sequencing depth. Why were neutrophils excluded from the "Old-Keto and Old-Chow dataset"? How were the doublets detected? (v) Why was a % of mitochondrial genes set at 5%, as it is possible that highly activated cells have higher expression of mitochondrial genes? Does the KD have an effect on mitochondrial gene expression?

(i) Even though we did not perform scRNA-seq with uninfected mice, based on Reviewer’s suggestions, we have re-analyzed the scRNA-seq data in young and old infected mice, focusing on responses to infection associated with aging. We identified three monocyte populations in young and old chow-fed infected mice that match populations described in Figures 6F-H. Aging was mostly associated with a decrease in the *Ifi44*-expressing monocyte subpopulation (cluster 0), while cluster 1, affected the most by ketogenic diet, was not subject to age-associated changes. These data are included in the Figure 6—figure supplement 2I-K. We also described the data in Results section (Line 296-298).

(ii) We didn’t perform scRNA-seq with uninfected young and old mice in this study. We have performed these analyses in a separate study and posted the data “Dysregulation of adipose ILC2 underlies thermogenic failure in aging | bioRxiv” Indeed, as predicted by the Reviewer, there are dramatic changes in tissue-resident immune system in aging. We believe these analyses will detract from the main point of the paper that ketogenic diet in aging protects against coronavirus infection.

(iii) Results of per cell type differential expression analysis between old infected chow- and KD-fed mice are summarized in Figure 6—figure supplement 2 panel B. While population structure was substantially affected by ketogenic diet (Figure 6B), we observed only very modest changes within few clusters. No shared signature was found. We included the description in Results section of manuscript (Line 267-269).

(iv) Numbers of cells used in the analysis are specified for each sample in Figure 6A and Figure 6—figure supplement 1 panel A. Quality control parameters, including sequencing depth, are summarized in Author response image 1.

Doublets were manually defined as cells that were positive for two or more marker sets (e.g. highly expressing both Cd3e (T cell) and Lyz2 (myeloid)). Corresponding clarification has been added to the methods section (Line 569-570). Neutrophils have extremely low RNA content, and generation of representative single-cell data for this cell type, as well as for other granulocytes, remains challenging (https://kb.10xgenomics.com/hc/en-us/articles/360004024032-Can-I-process-neutrophils-or-other-granulocytes-using-10x-Single-Cell-applications- ). Accordingly, neutrophils and red blood cells demonstrated very low gene counts in our data. Additionally, we previously observed that estimation of neutrophil abundances based on single-cell RNA-seq data cannot always be reproduced using extracellular markers (Gubin et al., 2018, PMID: 30445041).(v) Fraction of reads mapping to mitochondrial genes (mtDNA) is a widely used quality control metric to filter out dead cells. In a cell with disrupted membranes, cytosolic RNA will leak out while mitochondria will stay within the cell. Therefore, the fraction of mitochondrial genes will be increased in dying cells (Osorio et al., 2020, PMID: 32840568). 5% is a standard threshold accepted by the field, as per 10x technical note (https://assets.ctfassets.net/an68im79xiti/4tVumiyINGgAeoCg8SiWGG/1cf0888200d668142612c8d3f3679cf4/CG000130_10x_Technical_Note_DeadCell_Removal_RevA.pdf ) samples with ~80% viable cells have fraction of mitochondrial UMI around 3%. Per cluster differential expression analysis comparing old chow- and KD-fed mice identified only three mitochondrial genes that change with ketogenic feeding in at least one cell type. No coordinated increase in mitochondrial genes or pathways was observed. We have included these details in Author response image 2.

**Author response image 2. respfig2:** 

12. Did the authors also analyze with single cell-RNA sequencing of infected young mice? Do those mice also miss cluster 1 monocyte population (referring to Figure 6F)? This would imply the relative importance of those cells in the immunopathology of the lungs in aged mice. Further, would it be possible to characterize the monocytes from cluster 1 with extracellular markers within the flow cytometry data?

As discussed above (11i), based on Reviewer’s suggestions, we have re-analyzed the scRNA-seq data. We identified three monocyte populations in young and old chow-fed infected mice that match populations described in Figures 6F-H. Aging was mostly associated with a decrease in *Ifi44*-expressing monocyte subpopulation (cluster 0), while cluster 1, affected the most by ketogenic diet, was not subject to age-associated changes. These data are included in the Figure 6—figure supplement 2I-K. We also described the data in Results section (Line 296-298).